# Severe mental illness and health service utilisation for nonpsychiatric medical disorders: A systematic review and meta-analysis

Amy Ronaldson[1,2]*, Lotte Elton[1], Simone Jayakumar[1], Anna Jieman[1], Kristoffer Halvorsrud[1], Kamaldeep Bhui[1,3]

**1** Centre for Psychiatry, Wolfson Institute of Preventive Medicine, Barts & The London School of Medicine, Queen Mary University of London, London, United Kingdom, **2** Department of Psychological Medicine, Institute of Psychiatry, Psychology and Neuroscience, King's College London, London, United Kingdom, **3** Department of Psychiatry, University of Oxford, Oxford, United Kingdom

* amy.ronaldson@kcl.ac.uk

## Abstract

**Data Availability Statement:** As this is a systematic review of existing literature all relevant

### Background

Psychiatric comorbidity is known to impact upon use of nonpsychiatric health services. The aim of this systematic review and meta-analysis was to assess the specific impact of severe mental illness (SMI) on the use of inpatient, emergency, and primary care services for nonpsychiatric medical disorders.

### Methods and findings

PubMed, Web of Science, PsychINFO, EMBASE, and The Cochrane Library were searched for relevant studies up to October 2018. An updated search was carried out up to the end of February 2020. Studies were included if they assessed the impact of SMI on nonpsychiatric inpatient, emergency, and primary care service use in adults. Study designs eligible for review included observational cohort and case-control studies and randomised controlled trials. Random-effects meta-analyses of the effect of SMI on inpatient admissions, length of hospital stay, 30-day hospital readmission rates, and emergency department use were performed. This review protocol is registered in PROSPERO (CRD420191 19516). Seventy-four studies were eligible for review. All were observational cohort or case-control studies carried out in high-income countries. Sample sizes ranged from 27 to 10,777,210. Study quality was assessed using the Newcastle-Ottawa Scale for observational studies. The majority of studies ($n = 45$) were deemed to be of good quality. Narrative analysis showed that SMI led to increases in use of inpatient, emergency, and primary care services. Meta-analyses revealed that patients with SMI were more likely to be admitted as nonpsychiatric inpatients (pooled odds ratio [OR] = 1.84, 95% confidence interval [CI] 1.21–2.80, $p = 0.005$, $I^2 = 100\%$), had hospital stays that were increased by 0.59 days (pooled standardised mean difference = 0.59 days, 95% CI 0.36–0.83, $p < 0.001$, $I^2 = 100\%$), were

data are within the manuscript and its Supporting Information files.

**Funding:** There was no funding source for the study.

**Competing interests:** The authors have declared that no competing interests exist.

**Abbreviations:** ACG, Adjusted Clinical Groups; ACS, acute coronary syndrome; ACSC, ambulatory care sensitive condition; ADL, Activities of Daily Living; AHRQ, Agency for Healthcare Research and Quality; AIDS, acquired immune deficiency syndrome; APR-DRG, All-Patient Refined Diagnostic Related Groups; ASA, American Society of Anesthesiologists; BMI, body mass index; CI, confidence interval; CMHCB-DP, Case Management for High-Cost Beneficiaries Demonstration Project; COPD, chronic obstructive pulmonary disorder; DSM-IV, Diagnostic and Statistical Manual of Mental Disorders-IV; DVT, deep vein thrombosis; EU, European Union; GP, general practitioner; HCC, hierarchical condition category; HCUP-NIS, Healthcare Cost and Utilization Project–National Inpatient Sample; HCUP SID, Healthcare Cost and Utilization Project Stat Inpatient Database; HIV, human immunodeficiency virus; HPA, hypothalamic pituitary adrenal; IBD, inflammatory bowel disease; ICD, International Statistical Classification of Diseases and Related Health Problems; IMS, Intercontinental Marketing Services; IRR, incidence rate ratio; LOS, length of stay; MDD, major depressive disorder; NIHSS, National Institute for Health Stroke Scale; NOS, Newcastle-Ottawa Scale; NR, not reported; OR, odds ratio; PAD, peripheral artery disease; PPH, potentially preventable hospitalisation; PRISMA, Preferred Reporting Items for Systematic Reviews and Meta-Analysis; PTSD, posttraumatic stress disorder; PVD, peripheral vascular disease; QOF, Quality Outcomes Framework; RR, risk ratio; RTT, return to theatre; SMD, standardised mean difference; SMI, severe mental illness; TBSA, total burn surface area; UHC, University Health System Consortium; VA, Veteran's Association; VASQIP, VA Quality Improvement Program.

more likely to be readmitted to hospital within 30 days (pooled OR = 1.37, 95% CI 1.28–1.47, $p < 0.001$, $I^2 = 83\%$), and were more likely to attend the emergency department (pooled OR = 1.97, 95% CI 1.41–2.76, $p < 0.001$, $I^2 = 99\%$) compared to patients without SMI. Study limitations include considerable heterogeneity across studies, meaning that results of meta-analyses should be interpreted with caution, and the fact that it was not always possible to determine whether service use outcomes definitively excluded mental health treatment.

## Conclusions

In this study, we found that SMI impacts significantly upon the use of nonpsychiatric health services. Illustrating and quantifying this helps to build a case for and guide the delivery of system-wide integration of mental and physical health services.

---

## Author summary

### Why was this study done?

- The evidence to date suggests that psychiatric comorbidity leads to increased utilisation of general medical services.

- The literature assessing the impact of psychiatric comorbidity overall and of common mental health disorders on the use of nonpsychiatric health service use has been systematically reviewed. However, to date, the literature surrounding severe mental illness (SMI) and nonpsychiatric health service utilisation is yet to be reviewed.

### What did the researchers do and find?

- This systematic review incorporates the findings of 74 studies (all observational cohort or case-control studies) that reported on the impact of SMI on inpatient hospital admissions, length of hospital stay, 30-day readmission rates, emergency department visits, and use of primary care services.

- Narrative synthesis and random-effects meta-analyses showed that having SMI is associated with increased utilisation of all health services included as outcomes in the review.

- Large amounts of variation between studies in terms of patient population and health systems means that the results of the meta-analyses should be interpreted with caution.

### What do these findings mean?

- The results of this review highlight the extent to which SMI impacts upon nonpsychiatric health service utilisation.

- Illustrating and quantifying this helps to build a case for system-wide integration of mental and physical healthcare.

## Introduction

Mental health conditions are associated with high disease burden, poor overall health outcomes, and high health service utilisation [1–4]. Arguably, increased health service utilisation could be attributed to appropriate use of psychological and psychiatric services. However, an early review (1994) of the literature found that psychiatric comorbidity (particularly cognitive and organic mental disorders) was associated with increased length of stay (LOS) in the general hospital [5]. Building on this, a later review (2005) assessed nonorganic common mental disorders and found that depression was associated with higher use of general medical services [6]. The most recent review (2018) found that medical inpatients with any psychiatric comorbidity had longer hospital stays, higher medical costs, and more readmissions than inpatients without [7].

Although severe mental illness (SMI) was not precluded from Jansen and colleagues' review [7], the specific impact of SMI on nonpsychiatric health service use was not reviewed or quantified. People with SMI, such as schizophrenia or psychotic disorder, are more likely to develop chronic physical illness than the general population [8], and the impact of physical illness on people with SMI is significantly greater [9]. It is probable that this affects the use of nonpsychiatric general medical services in this patient group. Moreover, there are serious inequalities in the provision of physical healthcare for patients with SMI [8–10], which likely have repercussions for how they use general medical services.

Therefore, we sought to specifically review the literature surrounding the impact of SMI on the use of nonpsychiatric inpatient, emergency, and primary care services for patients with medical disorders. When possible, meta-analysis was used to determine the effect that SMI had on specific outcomes.

## Methods

This review protocol is registered in the PROSPERO International Prospective Register of Systematic Reviews (https://www.crd.york.ac.uk/PROSPERO/) (CRD42019119516). The protocol conforms to the Preferred Reporting Items for Systematic Reviews and Meta-Analysis (PRISMA) guidelines; the relevant checklist is provided in S1 Appendix. This research is part of a larger systematic review assessing the impact of SMI and personality disorder on nonpsychiatric health service utilisation. The present study focused on the impact of SMI on inpatient, emergency, and primary care service use. Ethical approval was not required.

### Search strategy and selection criteria

We searched PUBMED, Web of Science, PsycINFO, EMBASE, and The Cochrane Library for relevant studies with no publication date restrictions (see S2 Appendix for the full search strategy for each database searched). The search was supplemented with hand searches of journals related to the field and reference sections of relevant papers. Searches were carried out between 26 October 2018 and 2 November 2018. The search strategy was developed and conducted by AR. An updated search was carried out up to the end of February 2020.

Different definitions of SMI exist, but it generally refers to illnesses associated with psychosis as adopted by the Quality and Outcomes Framework of the United Kingdom (UK) National Health Service [11]. Therefore, in this review, SMI included bipolar disorder, psychosis, schizophrenia, and/or schizoaffective disorder. SMI did not include major depressive disorder (MDD). Studies that included MDD in their definition of SMI were excluded unless results were presented separately for each SMI subtype. Patients with physical health conditions and/or receiving medical treatment who did not have SMI served as controls.

Health service utilisation for medical nonpsychiatric disorders was the primary outcome. The current paper focused on inpatient (number of hospital admissions, likelihood of hospital admission, LOS in days, risk of longer LOS, number of readmissions, likelihood of readmissions), emergency (number of emergency department visits, likelihood of an emergency department visit), and primary care service use (number of primary care visits, likelihood of a primary care visit). We excluded studies for service use related to psychiatric, psychological, mental, or behavioural disorders. We included observational cohort and case-control studies and randomised controlled trials. We excluded reviews, case reports, and studies that used qualitative methods only.

For a study to be included, patients had to be 16 years or older. The majority of studies were explicit about excluding paediatric/adolescent patients. Where it was unstated whether all patients were over 16 years (four studies), the decision to include the study was based on the likelihood of the index medical condition occurring in a paediatric sample. For example, a sample of stroke patients will unlikely have a sizeable paediatric sample and would therefore be included, whereas a sample of patients undergoing trauma surgery could have a sizeable paediatric sample and would be excluded. All studies had to be published in peer-reviewed journals. Non-English-language articles were excluded. Conference proceedings were also excluded. When conference proceedings emerged in the search, authors were contacted to ascertain whether the data had been published in a peer-reviewed journal.

Articles were independently screened in two stages: a title and abstract screen (AR, AJ), followed by the retrieval and screening of potentially relevant full-text articles by two reviewers using the criteria listed above (AR, LE). Interrater reliability for the full-text screen was assessed using Cohen's kappa, which indicated moderate and substantial levels of agreement between the reviewers for the original and updated search, respectively (original: κ 0.55, 77.6% agreement; updated: κ 0.64, 81.8% agreement). Conflicts were resolved through discussion.

## Data extraction and quality assessment

Data were extracted by two reviewers (AR, SJ): AR extracted the data from the publications and SJ cross-checked 10% of the extracted studies for accuracy. There was acceptable agreement on extraction (intraclass correlation coefficient = 0.63). Sample characteristics, methodological characteristics, and main health service utilisation outcomes were extracted. The data extraction tables were piloted and refined before extraction began. As all studies included in the review were observational studies, the Newcastle-Ottawa Scale (NOS) was used to assess the quality of each study [12]. The NOS assesses the quality of each study using a system in which 'stars' are awarded on three broad categories: selection of groups, comparability of groups, and discernment of the outcome of interest for the case-control or cohort. Each article is rated on nine variables and can earn a maximum of nine stars. More stars indicate less risk of bias in a given study, and the number of stars awarded allows a study to be deemed of good, fair, or poor quality. Because of the nature of the current review, if a study did not adjust for severity of physical illness and/or the presence of physical comorbidities (e.g., Charlson Comorbidity Index [13], Elixhauser Comorbidity Index [14], a list of relevant physical comorbidities, a measure of physical illness severity), it was deemed to be of poor quality, regardless of the number of NOS 'stars' acquired. Quality assessment was carried out with the outcome of interest in mind; i.e., if a study had several clinical outcomes alongside a health service use outcome, the quality of the study would be assessed based on the health service use outcome.

## Data analysis

For all outcomes (inpatient service use, emergency service use, and primary care use), a narrative synthesis was carried out.

Meta-analysis and subgroup analysis were performed using Review Manager 5.3 of the Cochrane Collaboration [15]. In all meta-analyses, we used a random-effects model, since this model estimates effects while considering the heterogeneity between studies. For studies that reported continuous data, only those that provided both means and standard deviations/standard errors were included in the meta-analysis. Where required, standard deviations were calculated from confidence intervals (CIs) using a verified formula [16]. For studies that reported the likelihood of an outcome occurring, only adjusted studies (physical comorbidities/illness severity + other relevant factors) that provided odds ratios (ORs) and 95% CIs were included in the meta-analysis. All ORs and CI limits were log-transformed (natural log). Standard error was then calculated using a verified formula [16].

Higgins' $I^2$ was used to assess heterogeneity between studies. Where considerable heterogeneity is present ($I^2 \geq 90\%$) [16], statistical pooling is usually deemed inappropriate. However, we have reported pooled effect sizes for ease of interpretation, even in cases where there was considerable heterogeneity due to the clinical relevance of results. These should be interpreted with caution.

Sources of heterogeneity were investigated using subgroup analysis. Sources investigated included type of health service use outcome (all-cause [i.e., psychiatric treatment is unlikely but cannot be definitively ruled out] versus medical), type of SMI, sample size (cutoff determined by median split), country where the study took place, and study quality, when applicable. We assessed the degree of publication bias by visual examination of funnel plots. Publication bias was deemed to be absent if the plot showed an inverted symmetrical funnel. In all analyses, statistical significance was set at $p \leq 0.05$.

## Results

### Study selection

The systematic literature search resulted in a total of 4,620 articles. After removing duplicates, 3,507 articles remained. Preliminary hand searches of journals and reference sections of relevant papers identified eight more articles. After reviewing the titles and abstracts of these 3,515 articles, a total of 196 articles were included for the full-text review. The updated search carried out in March 2020 resulted in a total of 347 articles. After reviewing the titles and abstracts of these 347 articles, 33 were included in the full-text review (see PRISMA diagram in Fig 1). A list of excluded articles is provided in S3 Appendix.

Of 87 eligible studies, 74 (initial search, 61 studies; updated search, 13 studies) entered the review, as these assessed the impact of SMI on inpatient, emergency, and/or primary care services. All were observational cohort or case-control studies, and most were retrospective cohorts in which health service utilisation was compared between patients with and without SMI over time ($n = 63$); some studies adopted matched case-control designs ($n = 11$). Forty-five studies were deemed to be of good quality, three were fair, and 26 were poor.

The number of participants in the reviewed studies ranged from 27 to 10,777,210. The majority of the studies were from the United States (US: $n = 51$), with the remainder in the UK ($n = 7$), Canada ($n = 4$), Denmark ($n = 4$), Taiwan ($n = 3$), Australia ($n = 2$), Israel ($n = 1$), Japan ($n = 1$), and Sweden ($n = 1$). The majority of studies were carried out in patients with specific medical index disorders ($n = 54$). The most common medical index disorders were diabetes ($n = 10$), heart failure ($n = 7$), and total joint (knee and/or hip) arthroplasty ($n = 7$). The remaining studies ($n = 20$) were carried out in the general medical population, e.g., patients admitted to general medicine departments [17,18], all residents of nursing homes in Florida [19], all people on the Taiwanese National Health Research Institute Database [20]. Study periods ranged from 3 months to 18 years.

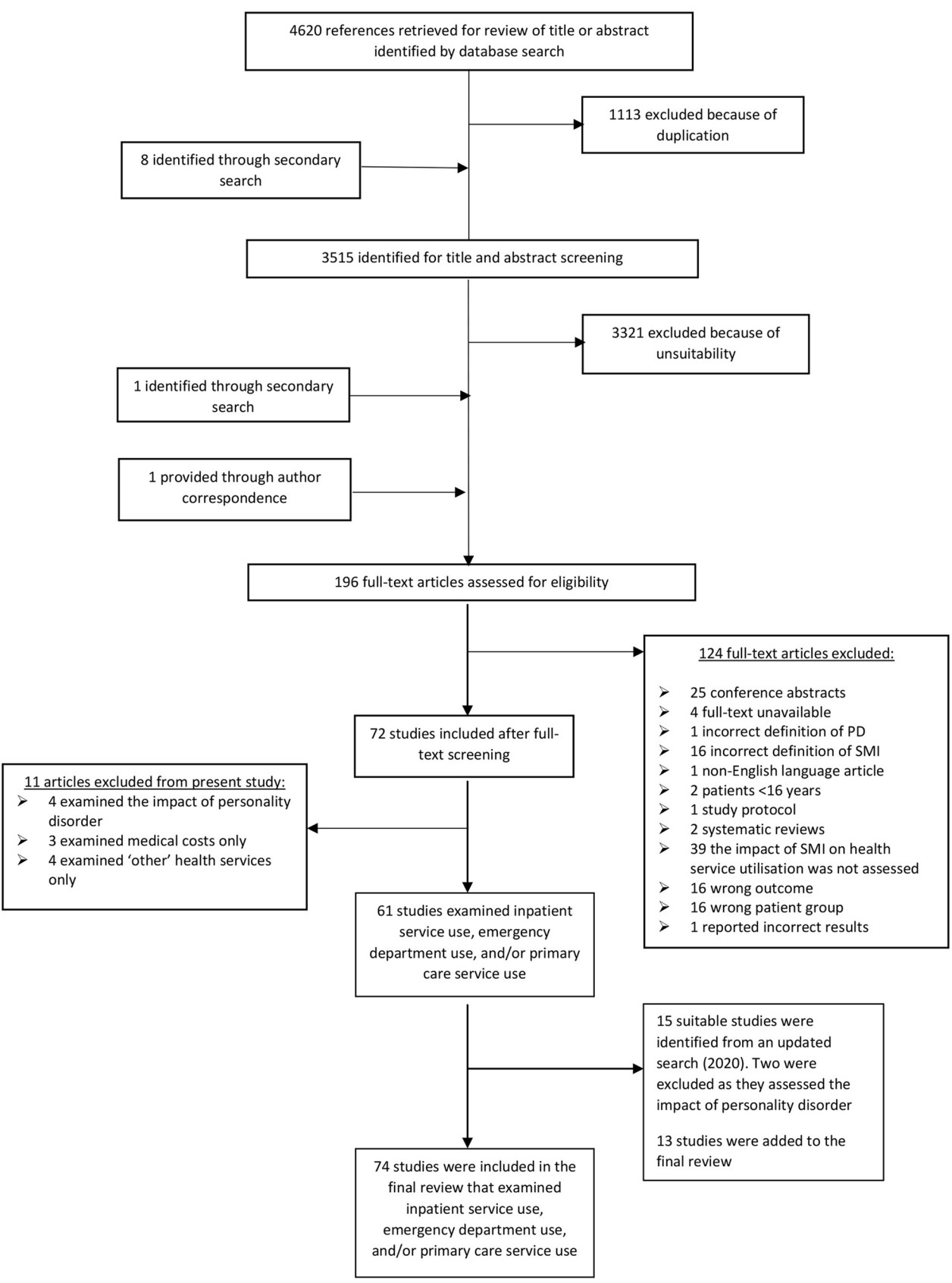

**Fig 1. PRISMA flowchart of study selection.** PRISMA, Preferred Reporting Items for Systematic Reviews and Meta-Analysis; SMI, severe mental illness.

### The impact of SMI on the use of nonpsychiatric medical inpatient services

**Inpatient hospital admissions.** Table 1 describes the 27 studies (comprising 44 separate analyses) of the impact of SMI on nonpsychiatric medical inpatient admissions [19–45]. In 35 of these analyses, having SMI was associated with increased inpatient hospital admissions over the study period (12 months to 15.5 years). Seven analyses from five studies produced nonsignificant results [22,28,33,35,43] and two analyses revealed that major psychotic disorder and schizophrenia was associated with reduced inpatient admissions in residents of assisted living facilities under 65 years and veterans under 60 years of age, respectively [22,33].

Results from 17 analyses from nine studies were deemed potentially appropriate for inclusion in a meta-analysis [21,23,25,27,31,33,35,36,43]; i.e., they adjusted for physical comorbidities/illness severity and other relevant factors and presented ORs relating to the likelihood of inpatient hospitalisation over the study period. However, Higgins' $I^2$ indicated that the set of studies were extremely heterogeneous ($\chi^2$ [16] = 10,649.77, $p < 0.001$, $I^2$ = 100%); therefore, the estimation of the overall pooled effect should be interpreted with caution. The pooled OR indicated that patients with SMI were significantly more likely to be admitted as a nonpsychiatric inpatient than patients without SMI (pooled OR = 1.84, 95% CI 1.21–2.80, $p$ = 0.005; Fig 2).

In order to determine the source of heterogeneity, we examined estimates between analyses that examined all-cause hospital admissions versus medical admissions, SMI subtype, sample size (determined by median split; $n > 155,312$ versus $n \leq 155,312$), and analyses in the US versus Canada. Results showed that likelihood of inpatient admission did differ across SMI subtype (test for subgroup differences: $\chi^2$[2] = 11.35, $p$ = 0.003) with patients with bipolar disorder having the lowest risk of admission (OR = 1.35, 95% CI 1.07–1.71, $p$ = 0.01) and those with schizophrenia having the highest (OR = 2.37, 95% CI 1.09–5.15, $p$ = 0.03). No other factors explained heterogeneity (see S4 Appendix for tables detailing subgroup analyses statistics). Study quality was not considered to be a potential source of heterogeneity, as all but one study included in the meta-analysis was of good quality. Visual examination of the funnel plot (see S5 Appendix) did not indicate significant publication bias.

**Length of hospital stay.** Table 2 describes the 30 studies (containing 38 separate analyses) that assessed the impact of SMI on nonpsychiatric LOS [17,18,20,28–30,32,40–42,46–65]. In 29 of the 38 analyses, SMI was associated with increased LOS. Eight analyses reported no significant associations [46,48,50,51,63,65], and one study found SMI associated with shorter hospital stays [29].

Fifteen studies (17 analyses) were potentially suitable for meta-analysis, as they reported the mean LOS (and standard deviation or CIs) for patients with and without SMI [20,28,30,41,42, 46,48,51,54,56,58,59,61,62,65]. The Higgins' $I^2$ indicated that the set of studies were extremely heterogeneous ($\chi^2$[16] = 15,432.12, $p < 0.001$, $I^2$ = 100%); therefore, the estimation of the overall pooled effect should be interpreted with caution. The pooled standardised mean difference (SMD) indicated that nonpsychiatric LOS was 0.59 days longer for patients with SMI compared to patients without (pooled SMD = 0.59 days, 95% CI 0.36–0.83, $p < 0.001$; Fig 3).

Sources of potential heterogeneity were investigated using subgroup analysis (see S4 Appendix). Results indicated that LOS differed across SMI subtypes (test for subgroup differences: $\chi^2$[3] = 15.73, $p$ = 0.001), with patients with bipolar disorder having the shortest difference in LOS (SMD = 0.11 days, 95% CI −0.03 to 0.25) and those with schizophrenia having the highest (SMD = 0.86 days, 95% CI 0.50–1.21). There was also a difference in terms of study

**Table 1. The impact of comorbid SMI on inpatient admissions.**

| Authors | Population (n) | Comorbid SMI (n) | SMI assessment | Type of admission | Control for comorbidities/medical illness severity | Control for other variables | Main findings | NOS |
|---|---|---|---|---|---|---|---|---|
| Basta et al. 2016 [21] | All women who underwent lumpectomy or mastectomy with a concurrent axillary lymph node procedure for a diagnosis of breast cancer (n = 56,075) | Psychosis (n = NR) | Elixhauser Comorbidity Index | Complicated lymphedema inpatient admission within 2 years of initial surgical procedure | Elixhauser Comorbidity Index, cancer in situ versus invasive cancer, type of surgery | Age, insurance type, tobacco use | Psychosis was associated with an increased risk of complicated lymphedema inpatient admission within 2 years of surgery (OR = 2.15, 95% CI 1.51–3.06) | 8 Good quality |
| Becker et al. 2010 [19] | Al residents of nursing homes in Florida (n = 72,251) | Bipolar disorder (n = 657) Major psychotic disorder (n = 10,141) | ICD-9 diagnostic codes from Medicaid inpatient claims | Number of hospitalisations for ACSGs over 3-year study period | Charlson Comorbidity Index, diagnosis of dementia, alcohol-use disorder, drug-use disorder | Age, gender, race | Bipolar disorder was associated with an increased likelihood of hospitalisation for ACSCs during the study period (HR = 1.79, 95% CI 1.50–2.15, p < 0.001) Major psychotic disorder was associated with an increased likelihood of hospitalisation for ACSCs during the study period (HR = 1.44, 95% CI 1.36–1.52, p < 0.001) | 7 Good quality |
| Becker et al. 2012 [22] | All residents of assisted living facilities in Florida (n = 16,208) Separate analyses carried out for those under the age of 65 years and those 65 years or older (n = 7,991) | Bipolar disorder (<65 years old: n = 215, ≥65 years: n = 139) Major psychotic disorder (<65 years old: n = 6,953, ≥65 years: n = 2,132) | ICD-9 diagnostic codes from Medicaid inpatient claims | Number of hospitalisations for ACSCs over 5-year study period | Charlson Comorbidity Index, diagnosis of dementia, alcohol-use disorder, drug-use disorder | Age, gender, race | In people under 65 years, bipolar disorder was not associated with an increased likelihood of hospitalisation for ACSCs during the study period (HR = 1.15, 95% CI 0.75–1.76, p = 0.534). Major psychotic disorder was associated with a reduced likelihood of hospitalisation (HR = 0.69, 95% CI 0.58–0.84, p < 0.001) In people 65 years or older, neither bipolar disorder (HR = 0.87,95% CI 0.58–1.31, p = 0.433) nor major psychotic disorder (HR = 0.93, 95% CI 0.82–1.05, p = 0.238) was associated with an increased likelihood of hospitalisation for ACSCs | 7 Good quality |
| Bresee et al. 2012 [23] | All patients on the Administrative Database of Alberta Health and Wellness (1995–2006) (n = 2,310,391) | Schizophrenia (n = 28,755) | ICD-9 and ICD-10 diagnostic codes from physician claims data and hospital discharge data | Hospitalisation (for any reason) over the 2-year study period; ≥1 yearly hospitalisation | Diagnosis of coronary artery disease and/or diabetes | Age, sex, SES, urban versus rural dwelling | Patients with schizophrenia were more likely to be hospitalised for any reason over the study period (OR = 5.67, 95% CI 5.50–5.84) and were more likely to have one or more hospitalisations per year (OR = 7.88, 95% CI 7.54–8.23) | 8 Good quality |
| Chen et al. 2007 [24] | Patients with Parkinson's disease (n = 43,772) | Psychosis (n = 4,429) | Not specified | Risk of medical hospitalisation over a 12-month period | No | Age, race, marital status | Psychosis was associated with an increased risk of medical hospitalisations over a 12-month period in patients with Parkinson's disease (OR = 1.75, 95% CI 1.61–1.90, p < 0.001) | 6 Poor quality* |
| Cramer et al. 2010 [25] | Patients with diabetes with at least two hospitalisations in the study period (12 months) (n = 695) | Psychosis (n = NR) | ICD-9 diagnostic codes from Medicaid claims data | Number of inpatient admissions during the 12-month study period | Diabetic acute complications, lower extremity disease, ischaemic heart disease, congestive heart failure, stroke, cardiac arrhythmia, COPD, drug/alcohol/substance abuse | Gender | Psychosis was associated with an increased risk of inpatient admissions over a 12-month period among diabetes patients with multiple hospitalisations (OR = 2.15, 95% CI 1.18–3.92) | 6 Fair quality |
| Davydow et al. 2016 [26] | Patients hospitalised for ACSCs (n = 5,945,540) | Bipolar disorder (n = 25,648) Schizophrenia (n = 42,558) | Diagnosis recorded in the Danish Civil Registration System | Hospitalisations for ACSCs during the 15-year study | Charlson Comorbidity Index | Age, sex, calendar period, marital status, education level, substance abuse, primary care use | Patients with SMI (bipolar disorder and/or schizophrenia) were at increased risk of hospitalisation for all chronic and acute ACSCs (IRR = 1.41, 95% CI 1.37–1.45, p < 0.001) When looked at separately, both patients with schizophrenia (IRR = 1.47, 95% CI 1.41–1.54, p < 0.001) and bipolar disorder (IRR = 1.33, 95% CI 1.27–1.39, p < 0.001) had increased risk for ACSCs hospitalisations also | 9 Good quality |
| Graham et al. 2019 [27] | Patients undergoing noncardiac inpatient surgery assessed by the VASQIP (n = 280,681) | Psychosis (n = 5,867); bipolar disorder (n = 6,337) | ICD-9 diagnostic codes | Inpatient admissions for any reason 24 months before and 24 months after the surgical procedure | Charlson Comorbidity Index | Age, sex, marital status, BMI, previous ED visits, discharge location after surgery, surgical specialty, smoking status, emergency case status, postoperative diagnosis, functional status | Compared to those who were independent in terms of functional status, those with bipolar disorder (OR = 1.95, 95% CI 1.79–2.12, p < 0.0001) and psychosis (OR = 2.05, 95% CI 1.89–2.23) had consistently higher healthcare utilisation (inpatient admissions pre- and postsurgery) | 7 Fair quality |
| Hendrie et al. 2014 [28] | Any patient receiving care at Wishard Health Services who were over or reached the age of 65 years during the study period (10 years) (n = 31,588) | Schizophrenia (n = 757) | ICD-9 diagnostic codes in electronic hospital records or in any of the linked datasets | Hospital admissions during the 10-year study period | A diagnosis of arthritis, coronary artery disease, cancer, heart failure, COPD, dementia, diabetes, alcohol abuse, hypertension, hyperthyroid, hypothyroid, liver disease, renal disease, stroke, substance abuse | Age, sex, race, smoking | In fully adjusted analysis, there was no difference between patients with schizophrenia and patients without schizophrenia in terms of admissions to hospital (p = 0.811) | 6 Good quality |
| Hsieh et al. 2012 [20] | Patients on the Taiwanese National Health Research Institute (n = 16,268) | Bipolar disorder (n = 4,067) | Acute admission ICD-9 diagnostic codes | Frequency of nonpsychiatric hospitalisations for those with bipolar disorder and matched controls over 2-year study period | No | Age, gender, urbanisation level of the residential area, monthly income | Patients with bipolar disorder (M = 0.37, SD = 1.11) had a significantly higher number of nonpsychiatric hospitalisations compared to matched controls (M = 0.15, SD = 0.48) over the study period (p < 0.001) | 7 Poor quality* |

(Continued)

**Table 1.** (Continued)

| Authors | Population (n) | Comorbid SMI (n) | SMI assessment | Type of admission | Control for comorbidities/medical illness severity | Control for other variables | Main findings | NOS |
|---|---|---|---|---|---|---|---|---|
| Hunter et al. 2015 [29] | Costliest 5% of Veterans Association patients (n = 261,515) | Bipolar disorder, schizophrenia, other psychosis (n = 33,119) | ICD-9 diagnostic codes and chronic condition indicators established by AHRQ | Number of hospitalisations (includes psychiatric care) over the 12-month study period | AHRQ comorbidity measures | Age, sex, race/ethnicity, marital status, documented homelessness during year of investigation, correlation within facilities | Having SMI was associated with a higher number of inpatient hospital admissions over the study period (p < 0.001). Patients with SMI had on average 2.2 admissions, whereas patients with no mental health conditions had on average 1.7 admissions | 8 Good quality |
| Krein et al. 2006 [30] | Patients with diabetes (n = 36,546) | Bipolar disorder, schizophrenia (excluding latent schizophrenia), schizoaffective disorder, other nonorganic psychoses, paranoid states, affective psychoses (n = 18,273) | ICD-9 diagnostic codes from National Psychosis Registry | Number of inpatient hospital stays over the 12-month study period | No | Patients matched on age | In unadjusted analyses, the likelihood of having an inpatient admission was increased for patients with SMI (OR = 2.80, 95% CI 2.67–2.94) | 6 Poor quality* |
| Kurdyak et al. 2017 [31] | Patients with diabetes (n = 1,131,375) | Schizophrenia (n = 26,259) | Ontario Health Insurance Plan records detailing three schizophrenia-related physician visits in 36 months, or a hospitalisation for schizophrenia | Number of hospitalisations for diabetic complications; number of hospitalisations for any nonmental health reason excluding trauma in the 2-year study period | Johns Hopkins ACG System, duration of diabetes | Age, sex, rural residence, neighbourhood income, neighbourhood material deprivation, past year service use | Patients with schizophrenia had an increased risk of hospitalisation for a diabetic complication (OR = 1.35, 95% CI 1.28–1.43) and an increased risk of hospitalisation for any nonmental health reason excluding trauma (OR = 1.85, 95% CI 1.79–1.92) | 8 Good quality |
| Lafeuille et al. 2014 [32] | Patients with substance dependence/abuse, obesity, diabetes, metabolic syndrome, hyperlipidaemia, hypertension, coronary artery disease, congestive heart failure, HIV, hepatitis C, or COPD (n = 49,304) | Schizophrenia (n = 24,652) | ICD-9 diagnostic codes—at least two primary or secondary schizophrenia diagnoses recorded during study period | Number of hospitalisations over the 10-year study period | Patients were matched 1:1 based on propensity scores using the Charlson Comorbidity Index | Patients were matched 1:1 based on propensity scores on age, gender, state, and year of index admission | Compared to matched nonschizophrenic controls, patients with schizophrenia had significantly higher rates of hospitalisations over the study period (IRR = 2.23, 95% CI 2.14–2.33, p < 0.005) | 8 Good quality |
| Lemke et al. 2012 [35] | All persons enrolled on the IMS Health Plan Claims Database for at least 6 months in 2006 (n = 4,632,226) | Bipolar disorder (approximately n = 23,160) | John Hopkins ACG system | Inpatient hospitalisations over the 2-year study period | Charlson Comorbidity Index, morbidity burden modelled using the John Hopkins ACG System in which a person is uniquely assigned to one of 106 groups; this model included 24 groups and 98 expanded diagnostic clusters | Age, sex, number of prior hospitalisations, ED episodes not resulting in inpatient hospitalisations, outpatient visits, markers for dialysis service, nursing services, and other major procedures | Patients with bipolar disorder of all ages who had experienced a prior hospitalisation in the base year had an increased risk of inpatient hospitalisation compared to those without (OR = 1.15, 95% CI 1.06–1.25). This was the same for patients with bipolar disorder aged ≥18 to <65 years who had no prior hospitalisation in the base year (OR = 1.68, 95% CI 1.58–1.79), but there was no significantly increased risk for patients with bipolar disorder, ≥65 years old, who had no prior hospitalisation in the base year (OR = 1.17, 95% CI 0.91–1.52) | 8 Good quality |
| Li et al. 2008 [36] | Patients with ACSCs (e.g., asthma, heart failure, hypertension) and patients with 'marker' conditions (e.g., appendicitis, acute myocardial infarction, hip/femur fracture). Conditions listed extensively in original paper (n = 155,312) | Bipolar disorder (n = 2,032) Schizophrenia (n = 2,714) Other psychoses (n = 795) | ICD-9 diagnostic codes from secondary diagnoses | Hospitalisations for ACSCs or 'marker' conditions over the 12-month study period | Elixhauser Comorbidity Index | Age, gender, race, insurance, HMO, income, tobacco use, admission type | Patients with bipolar disorder (OR = 1.87, 95% CI 1.53–2.30), schizophrenia (OR = 1.83, 95% CI 1.50–2.24), and other psychoses (OR = 2.64, 95% CI 1.78–3.90) all had increased likelihoods of being admitted to hospital for an ACSC or 'marker' condition within the study period | 7 Good quality |
| Lin et al. 2011 [37] | All patients included in the Longitudinal Health Insurance Database released by the Taiwan National Health Research Institute (n = 22,527) | Schizophrenia (n = 2,503) | ICD-9 diagnostic codes recorded as a principal diagnosis in the year prior to the study period | Number of hospitalisations for ACSCs (ruptured appendix, asthma, cellulitis, congestive heart failure, diabetes, gangrene, hypokalaemia, immunisable conditions, malignant hypertension, pneumonia, pyelonephritis, and perforated or bleeding ulcer) over the 5-year study period | No | Age, sex, level of urbanisation, geographic location, monthly income | Patients with schizophrenia had a 3.26-fold higher risk of ACSC hospitalisation than age and sex-matched controls (RR = 3.26, 95% CI 3.00–3.54, p < 0.001). When conditions known to be highly correlated with schizophrenia were removed from the analyses (asthma, diabetes, hypertension), patients with schizophrenia had a 2.46-fold higher risk of ACSC hospitalisations compared to the comparison group, after adjusting for potential confounders (RR = 2.46, 95% CI 2.12–2.86, p < 0.001) | 6 Poor quality* |
| Mai et al. 2011 [38] | People on the electoral roll (n = 433,388) The mental health cohort comprised people on both the electoral roll and the Mental Health Registry in Western Australia | Schizophrenia (approximately n = 5,847) Other psychoses (approximately n = 9,327) | ICD-9 diagnostic codes from the Mental Health Registry in Western Australia | Number of PPHs (ACSCs: vaccine-preventable, e.g., flu; chronic, e.g., diabetes complications; acute, e.g., appendicitis, and adverse drug events) over the 15.5-year study period | Charlson Comorbidity Index | Age, sex, indigenous status, level of social disadvantage, level of residential remoteness, year at start of follow-up | Compared to people with no mental health conditions, patients with schizophrenia (RR = 2.25, 95% CI 2.12–2.39) and patients with other psychoses (RR = 2.36, 95% CI 2.26–2.47) were more likely to have a PPH The leading cause of excess PPHs in patients with schizophrenia were nutritional deficiencies, adverse drug events, convulsions and epilepsy, congestive heart failure, and influenza and pneumonia The leading cause of excess PPHs in patients with other psychoses were convulsions and epilepsy, pyelonephritis, adverse drug events, other vaccine-preventable conditions, and gangrene | 8 Good quality |

(Continued)

**Table 1.** (Continued)

| Authors | Population (n) | Comorbid SMI (n) | SMI assessment | Type of admission | Control for comorbidities/medical illness severity | Control for other variables | Main findings | NOS |
|---|---|---|---|---|---|---|---|---|
| Minen et al. 2014 [39] | Patients who visited the ED with a primary diagnosis of migraine (n = 2,872) | Bipolar disorder (n = NR) | ICD-9 diagnostic codes recorded in the Partners Research Patient Data Registry, Massachusetts | Inpatient stays over the 10-year study period | No | No | Patients with bipolar disorder had 1.6 times more inpatient stays than patients with other psychiatric disorders. No inferential analyses were carried out on individual psychiatric conditions, and no comparison was made between patients with bipolar disorder and those without any psychiatric diagnosis | 7 Poor quality* |
| Moore et al. 2019 [33] | Veterans under 60 years of age from the Veterans Health Administration (n = 2,016,392) | Schizophrenia (n = 51,752); bipolar disorder (n = 77,839) | ICD-9 diagnostic codes | Medical-surgical hospital admissions over the study period (fiscal year 2012) | Charlson Comorbidity Index; seizures, cerebrovascular accident, chronic obstructive airway disease, peptic ulcer disease, hepatic disease, moderate/severe liver, pneumonia, nausea/vomiting, insomnia, any pain diagnosis | Homelessness, VA pension, service-connected 50% or more, tobacco use, opiate prescription | Veterans with schizophrenia had a lower risk of admission to medical-surgical hospital units than veterans with no mental health diagnoses (OR = 0.87, 95% CI 0.83–0.90). Veterans with bipolar disorder did not differ from veterans with no mental health diagnoses (OR = 1.02, 95% CI 0.99–1.06) | 8 Good quality |
| Norbeck et al. 2019 [34] | Homeless male veterans in urban and rural settings in the US (n = 156) | Bipolar disorder (n = 39) | Self-reported bipolar disorder | Self-reported use of inpatient services over the past 3 months | No | No | Having bipolar disorder was associated with an increase in overnight inpatient treatment (p = .034) | 4 Poor quality |
| Sayers et al. 2007 [40] | Patients ≥65 years old with congestive heart failure (n = 21,429) | Bipolar disorder (n = 58) Psychosis (n = 534) | Identified using AHRQ's Clinical Classifications software (ICD-9 diagnostic codes) | Estimated additional number of hospitalisations within the 12-month study period | Elixhauser Comorbidity Index | Age, sex, race, SES | Patients with bipolar disorder had an estimated 0.38 (30%) additional number of hospitalisations compared to patients with no psychiatric illness (p = 0.001). Patients with psychosis had an estimated 0.3 (24%) additional number of hospitalisations compared to patients with no psychiatric illness (p < 0.001) | 8 Good quality |
| Schoepf et al. 2014 [41] | Patients admitted for medical treatment across three hospitals (n = 15,598) | Schizophrenia (n = 1,418) | ICD-10 diagnostic codes | Number of hospital admissions over the 11.5-year study period study period | No | No | Patients with schizophrenia had almost 2-fold higher number of admissions to hospital within the study period (M = 11.5, SD = 0.4) compared to nonschizophrenic controls (M = 6.3, SD = 0.1) (p ≤ 0.001) | 7 Poor quality* |
| Sporinova et al. 2019 [42] | All adults (18+ years) in Alberta, Canada, with at least one of the following chronic diseases in 2012: asthma, congestive heart failure, myocardial infarction, diabetes, epilepsy, hypertension, chronic pulmonary disorder, and chronic kidney disease (n = 991,445) | Schizophrenia (n = 13,320) | ICD-10 diagnostic codes | Rate per 1,000 patient-days of chronic disease–specific hospitalisations over the 3-year study period | No | No | Patients with schizophrenia had a higher rate of chronic disease–specific hospitalisations (0.11, 0.10–0.12) compared to those without a mental health conditions (0.06, 0.06–0.06) | 7 Poor quality* |
| Sullivan et al. 2006 [43] | Patients visiting the ED with a primary diagnosis of diabetes (n = 4,275) | 'Psychotic' group which included patients with bipolar disorder and schizophrenia (n = 136) | ICD-9 diagnostic codes | Whether or not hospitalisation occurred at the end of the ED visit | Proxies for illness severity were time of arrival at the ED (11 PM–7 AM compared with all other times) and mode of arrival (ambulance versus other) | Age, gender, ethnicity | Patients with 'psychosis' were no more likely than patients with no mental illness to be hospitalised at the end of the ED visit (OR = 0.77, 95% CI 0.45–1.33) | 7 Good quality |
| Wallace et al. 2019 [44] | Medical claims data from the HealthCore Integrated Research Database (US) (n = 33,660) | Schizophrenia (n = 6,732) | ICD-9 and ICD-10 diagnostic codes | Rates of all-cause inpatient admissions in the year preceding schizophrenia diagnosis | No | Patients matched 1:4 on age, sex, and region of residence | Patients with schizophrenia had a higher rate of all-cause inpatient admissions (32.7%) compared to their matched comparators without schizophrenia (3.9%) | 7 Poor quality* |
| Wetmore et al. 2019 [45] | Patients on Medicare Claims Database who have Parkinson's disease (n = 52,103) | Psychosis (n = 2,778) | ICD-9 diagnostic codes | Number of inpatient admissions over the 6-year study period | Patients matched 1:4 based on number of comorbid conditions | Patients matched 1:4 on age, sex, race, index year of psychosis diagnosis | Patients with psychosis had more inpatient admissions (0.9) than patients without psychosis (0.5) in the sixth year of follow-up | 8 Good quality |

*If studies failed to adjust for physical comorbidities/illness severity in their analyses, they were deemed to be of poor quality regardless of 'stars' recorded using the NOS.

Abbreviations: ACG, Adjusted Clinical Groups; ACSC, ambulatory care sensitive condition; AHRQ, Agency for Healthcare Research and Quality; CI, confidence interval; COPD, chronic obstructive pulmonary disorder; ED, emergency department; HIV, human immunodeficiency virus; HMO, health maintenance organisation; HR, hazard ratio; ICD, International Statistical Classification of Diseases and Related Health Problems; IMS, Intercontinental Marketing Services; IRR, incidence rate ratio; M, mean; NOS, Newcastle-Ottawa Scale; NR, not reported; OR, odds ratio; PPH, potentially preventable hospitalisation; RR, risk ratio; SD, standard deviation; SES, socioeconomic status; SMI, severe mental illness; US, United States; VA, Veteran's Association; VASQIP, VA Quality Improvement Program.

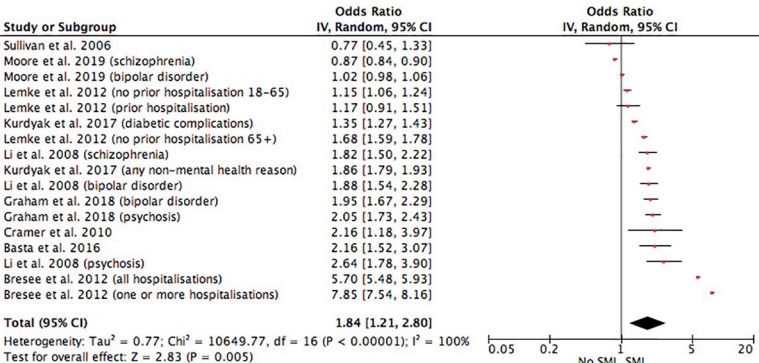

**Fig 2. The impact of SMI on inpatient hospital admissions.** Several studies included analyses for different samples and included different outcomes. Therefore, several effect estimates have been entered from the same studies. CI, confidence interval; SMI, severe mental illness.

quality with significant differences between those with and without SMI emerging only in studies of poor quality. No other sources of heterogeneity emerged (see S4 Appendix). Visual examination of the funnel plot (see S5 Appendix) suggested publication bias.

**Hospital readmission rates.** Table 3 describes the 20 studies (36 separate analyses) included in the review that examined the impact of SMI on nonpsychiatric hospital readmission rates [26,64,66–83]. In 32 of these analyses, SMI was associated with increased hospital readmissions. Four studies showed no significant association [70,72,74,75].

Ten studies (16 analyses) were suitable for meta-analysis [67–69,73,75,77–80,83]. These studies examined the impact of SMI on 30-day readmission rates specifically, except for one study that looked at 28-day readmission rates [75]. Higgins' $I^2$ ($\chi^2[15] = 86.25$, $p < 0.001$, $I^2 = 83\%$) suggested that meta-analysis was appropriate [16]. The pooled OR indicated that patients with SMI were significantly more likely than patients without SMI to be readmitted to hospital within 30 days of the index medical hospitalisation (pooled OR = 1.37, 95% CI 1.28–1.47, $p < 0.001$; Fig 4).

Although meta-analysis was appropriate, there was still considerable heterogeneity between studies examining 30-day readmission rates. Geographical area (test for subgroup differences: $\chi^2[1] = 13.89$, $p \leq 0.001$) was a significant sources of variance between studies (see S4 Appendix). Studies carried out in Europe (UK and Denmark) indicated that SMI patients were more likely to be readmitted to hospital within 30 days (OR = 1.65, 95% CI 1.48–1.85, $p < 0.001$) than their counterparts in the US (OR = 1.28, 95% CI 1.19–1.39, $p < 0.001$). No other sources of heterogeneity emerged. As all studies included in the meta-analysis were of good quality, study quality was not deemed to be a source of heterogeneity. Visual examination of the funnel plot (see S5 Appendix) indicated that publication bias was likely.

## The impact of SMI on emergency department use for medical disorders

Table 4 describes the 15 studies (20 analyses) included in the review that assessed the impact of SMI on the use of emergency care [20,23,29,31,32,34,39,42,44,45,70,76,84–86]. In 18 out of 20 of the analyses, SMI was associated with increased use of emergency departments. This was true irrespective of adjustments for severity of medical disorders. Only two studies reported nonsignificant differences in the use of the emergency department between patients with and without SMI [34,70].

Meta-analysis was suitable for four studies (six analyses) [23,31,84,86]. The Higgins $I^2$ value indicated that there was extreme variation between studies ($\chi^2[5] = 404.15$, $p < 0.001$, $I^2 =$

**Table 2. The impact of comorbid SMI on length of hospital stay.**

| Authors | Population (n) | Comorbid SMI (n) | SMI assessment | Type of LOS | Control for comorbidities/ medical illness severity | Control for other variables | Main findings | NOS |
|---|---|---|---|---|---|---|---|---|
| Attar et al. 2019 [46] | Patients in the Danish Register Study diagnosed with an ACS between 1995 and 2013 (n = 2,178) | Schizophrenia (n = 726) | ICD-10 diagnostic codes | LOS for ACS event (days) | Patients matched 1:2 with ACS patients without schizophrenia based on comorbidity risk score (hyperlipidaemia, obesity, atrial fibrillation, hypertension, heart failure, cardiomyopathy, sick sinus syndrome, valve disorder, diabetes, anaemia, COPD, PAD, stroke) | Patients matched 1:2 with ACS patients without schizophrenia based on sex, date of birth, year of ACS diagnosis | There was no statistically significant difference in LOS (days) between patients with schizophrenia (7.971 ± 9.41) and patients without schizophrenia (9.236 ± 19.33) ($p = 0.096$) | 9 Good quality |
| Bailey et al. 2018 [47] | Patients who underwent one of the four most common surgeries (cholecystectomy and common duct exploration, colorectal resection, excision and lysis of peritoneal adhesions, appendectomy) (n = 579,851) | Schizophrenia (n = 5,234) | DSM-IV diagnosis in patient records | Surgical LOS Likelihood of prolonged LOS, i.e., LOS greater than the 75th percentile (OR) | Elixhauser Comorbidity Index; surgery type; laparoscopic surgery; emergent or urgent surgery | Age, sex, race, payer type (US), median income quartile, hospital bed size, hospital ownership, hospital teaching status | Compared with patients who did not have a psychiatric diagnosis, patients with schizophrenia had an increased risk of prolonged length of hospital stay (OR = 1.64, 95% CI 1.52–1.78, $p < 0.001$) | 8 Good quality |
| Banta et al. 2010 [48] | Patients with congestive heart failure (n = 15,497) | Schizophrenia, PTSD, bipolar disorder (n = 564) | ICD-9 diagnostic codes recorded in patient notes in the 12 months prior to hospitalisation for heart failure | LOS for index hospitalisation for heart failure LOS in days (M, SD) | No | No | Compared with patients with no psychiatric comorbidity (M = 6.9 days, SD = 9.4 days), those with comorbid SMI only (i.e., no comorbid depression or anxiety) (M = 6.7 days, SD = 7.3 days) did not differ significantly in terms of LOS for their index hospitalisation for heart failure ($p = 0.62$)** | 6 Poor quality* |
| Beydoun et al. 2015 [49] | Population sample of patients with Alzheimer's disease (n = 99,260) | Psychoses (n = 6,652) | AHRQ comorbidity measure | Association between psychosis and increased LOS (beta) | AHRQ comorbidity measures | Age, sex, race, median household income, insurance status, admissions day (weekday versus weekend), bed size of hospital, ownership of hospital, location/ teaching status of hospital, region of the hospital | Having comorbid psychosis was associated with increased average LOS over the study period (12 months) (β = 2.05, SE = 0.15, $p < 0.001$) | 8 Good quality |
| Bressi et al. 2006 [17] | Patients hospitalised for medical conditions (n = 1,617,710) | Schizophrenia (n = NR) | ICD-9 diagnostic codes recorded as secondary diagnosis | LOS in days (M, no SD provided) | APR-DRG illness severity algorithm | Age, sex, race, payer type, region, discharge to long-term care, bed size of hospital, hospital location/teaching status | Patients with schizophrenia incurred hospital stays on average 0.86 days longer than patients without schizophrenia in the adjusted model, $p < .0001$ | 7 Good quality |
| Briskman et al. 2012 [18] | Patients hospitalised in general medicine departments (n = 192) | Bipolar disorder, schizophrenia, (n = 93) | Identified from medical records | Duration of hospitalisation (no M provided) | No | Age (matched controls) | The duration of hospitalisation was significantly longer for patients with comorbid bipolar disorder and/or schizophrenia compared to patients with no psychiatric diagnosis ($p < 0.035$). No descriptive statistics provided | 6 Poor quality* |

(*Continued*)

**Table 2.** (Continued)

| Authors | Population (n) | Comorbid SMI (n) | SMI assessment | Type of LOS | Control for comorbidities/ medical illness severity | Control for other variables | Main findings | NOS |
|---|---|---|---|---|---|---|---|---|
| Carter et al. 2016 [50] | Patients with heart failure (n = 31,760) | Bipolar disorder, schizophrenia (n = 237) | ICD-10 diagnostic codes recorded as secondary diagnosis | LOS in days (M, no SD, mean difference, CI) | No | No | LOS was significantly longer in patients with bipolar disorder (M = 20.4 days) compared to patients with no psychiatric disorder (M = 11.2 days) (mean difference = 8.8 days, 95% CI 3.5–14.2 days, p < 0.001) There was no difference in terms of LOS between patients with schizophrenia (M = 13 days) and patients with no psychiatric disorder (M = 11.2 days) (mean difference = 1.4 days, 95% CI −2.1 to 4.9) | 7 Poor quality* |
| Chen et al. 2019 [51] | Patients on the Taiwan National Health Insurance Database who had had a stroke during the 5-year study period (n = 2,320) | Bipolar disorder (n = 580) | ICD-9 diagnostic codes | LOS in days relating to admission for stroke | Patients were matched 1:3 with stroke patients without bipolar disorder on diagnoses of hypertension, hyperlipidaemia, diabetes, and CHD | Patients were matched 1:3 with stroke patients without bipolar disorder on age, sex, income, geographic location, urbanisation level of residence | LOS (days) did not differ between those with bipolar disorder (11.87 ± 16.11) and those without bipolar disorder (12.67 ± 16.51) (p = 0.27) | 9 Good quality |
| Cholankeril et al. 2016 [52] | Elderly patients hospitalised for colon cancer-directed surgery (n = 98,797) | Psychosis (n = 1,340) | ICD-9 diagnostic codes and HCUP-NIS comorbidity software | LOS mean difference in days | Elixhauser Comorbidity Index, surgical covariates (postoperative delirium, postoperative fistula or ileus, indication for reoperation, postoperative DVT, postoperative respiratory complications, intra-abdominal abscess) | Age, gender, ethnicity | Patients with psychosis had a longer hospital stay (mean difference = 6.4 days, p < 0.001) | 8 Good quality |
| Dolp et al. 2018 [53] | Burn patients with burns over 10% of total body surface area (n = 583) | Schizophrenia (n = NR) | NR | Likelihood of exceeding average LOS (OR) | Inhalation injury, %TBSA burn | Age, sex | Having comorbid schizophrenia increased the likelihood that a patient would exceed the average LOS (OR = 2.93, 95% CI 1.06–8.08) | 5 Fair quality |
| Falsgraf et al. 2017 [54] | Trauma patients (n = 26,502) | Bipolar disorder (approximately n = 212) Schizophrenia (approximately n = 344) | Inpatient hospital records | LOS in days (M, SD) | No | No | Patients with bipolar disorder (M = 10.1 days, SD = 14.7 days, p = 0.02) and patients with schizophrenia (M = 14.2 days, SD = 21.7 days, p < 0.001) experienced significantly longer LOS than patients with trauma and no psychiatric comorbidity (M = 6.2 days, SD = 14.4 days) | 7 Poor quality* |
| Gholson et al. 2018 [56] | Patients undergoing total joint arthroplasty (n = 505,840) | Schizophrenia (n = 953) | ICD-8 diagnostic codes | LOS in days (M, SD) | Charlson Comorbidity Index | Age; sex; smoking; race; use of corticosteroids; osteonecrosis of the hip; spasm of muscle; gait abnormality, contracture of joint, pelvic region, and thigh; vitamin D deficiency; surgery type | Patients with schizophrenia (M = 3.85 days, SD = 2.14 days) had a longer LOS than patients without schizophrenia (M = 3.22 days, SD = 1.32 days), p < 0.001 | 8 Good quality |
| Hendrie et al. 2014 [28] | Patients over 65 years old receiving care at Wishard Health Services, US (n = 31,588) | Schizophrenia (n = 757) | ICD-9 diagnostic codes in electronic hospital notes | LOS in days (M, SD) over the 10-year study period | No | No | Patients with schizophrenia had more total hospital days (M = 58.9 days, SD = 94.2 days) than patients without schizophrenia (M = 31.1 days, SD = 44.1 days), p < 0.001 | 6 Poor quality* |

(*Continued*)

**Table 2.** (Continued)

| Authors | Population (n) | Comorbid SMI (n) | SMI assessment | Type of LOS | Control for comorbidities/ medical illness severity | Control for other variables | Main findings | NOS |
|---|---|---|---|---|---|---|---|---|
| Hsieh et al. 2012 [20] | Patients on the Taiwanese National Health Research Institute (case-control design) (n = 16,268) | Bipolar disorder (n = 4,067) | Acute admission ICD-9 diagnostic codes | Inpatient LOS in days (M, SD) over the 2-year study period | No | Age, gender, urbanisation level of the residential area, monthly income | Over 2 years, patients with bipolar disorder (regression-adjusted annual M = 1.70 days, SD = 1.31 days) had longer inpatient LOS than matched controls without bipolar disorder (regression-adjusted annual M = 1.51, SD = 1.75), p < 0.001 | 7 Poor quality* |
| Hunter et al. 2015 [29] | Costliest 5% of Veterans Association patients (n = 261,515) | Bipolar disorder, schizophrenia, other psychosis (n = 33,119) | ICD-9 diagnostic codes and chronic condition indicators established by AHRQ | Medical-surgical LOS in days; long-term care LOS (M only) over the 12-month study period | AHRQ comorbidity measures | Age, sex, race/ethnicity, marital status, documented homelessness during year of investigation, correlation within facilities | Patients with SMI had longer long-term care LOS (M = 22.9 days) compared to patients with no mental health conditions (M = 11.4 days), p < 0.001 However, patients with SMI had shorter medical-surgical LOS (M = 6.4 days) compared to patients with no mental health conditions (M = 8.7 days), p < 0.001 | 8 Good quality |
| Kaplan et al. 2011 [55] | Patients with IBD who had undergone IBD-related surgery (n = 35,588) | Psychosis (n = 348) | Elixhauser Comorbidity Index | Association between psychosis and increased LOS (antilogarithms of regression coefficient presented to provide a percentage change in resource use) | Elixhauser Comorbidity Index | Age, sex, race, primary health insurer, emergency admission | Psychosis was associated with a 22% higher LOS (antilogarithm of regression coefficient = 1.22, 95% CI 1.16–1.29) | 8 Good quality |
| Krein et al. 2006 [30] | Patients with diabetes (n = 36,546) | Bipolar disorder, schizophrenia (excluding latent schizophrenia), schizoaffective disorder, other nonorganic psychoses, paranoid states, affective psychoses (n = 18,273) | ICD-9 diagnostic codes from National Psychosis Registry | LOS in days (M, SD) over the 12-month study period | No | Patients matched on age | Patients with SMI had longer mean LOS (M = 12.0 days, SD = 15.9 days) than patients without SMI (M = 8.2 days, SD = 11.4 days) | 6 Poor quality* |
| Lafeuille et al. 2014 [32] | Patients with substance dependence/abuse, obesity, diabetes, metabolic syndrome, hyperlipidaemia, hypertension, coronary artery disease, congestive heart failure, HIV, hepatitis C, or COPD (n = 49,304) | Schizophrenia (n = 24,652) | ICD-9 diagnostic codes—at least two primary or secondary schizophrenia diagnoses recorded during study period | Likelihood of prolonged LOS (not defined) (IRR) | Patients were matched 1:1 based on propensity scores using the Charlson Comorbidity Index | Patients were matched 1:1 based on propensity scores on age, gender, state, and year of index admission | Patients with schizophrenia had a significantly higher risk of prolonged LOS compared to nonschizophrenic matched controls (IRR = 2.17, 95% CI 1.90–2.47, p < 0.005) | 8 Good quality |
| Maeda et al. 2014 [57] | Patients undergoing major surgery (cardiovascular, intrathoracic, intraperitoneal, and suprainguinal-vascular procedures requiring general anaesthesia, excluding percutaneous procedures and obstetric surgery) (n = 5,569) | Schizophrenia (n = 104) | ICD-10 diagnostic codes from medical records | Association between psychosis and increased LOS (beta) | Charlson Comorbidity Index, delirium, intubation, haemodialysis related to surgery | Age, sex, surgery type, ambulance use, data year on clinical outcomes | Having schizophrenia was associated with an increased LOS (β = 0.48, 95% CI 0.32–0.64, p < 0.001) | 7 Good quality |
| Menendez et al. 2013 [58] | Patients with a lower extremity fracture (n = 10,669,449) | Schizophrenia (approximately n = 64,000) | ICD-9 diagnostic codes recorded in National Hospital Discharge Survey database | LOS in days (M, SD) | No | No | Patients with schizophrenia had a longer LOS (M = 11.0 days, SD = 21.0 days) compared to patients with no psychiatric diagnosis (M = 7.2 days, SD = 8.3 days), p < 0.001 | 7 Poor quality* |

(Continued)

**Table 2.** (Continued)

| Authors | Population (n) | Comorbid SMI (n) | SMI assessment | Type of LOS | Control for comorbidities/ medical illness severity | Control for other variables | Main findings | NOS |
|---------|---------------|------------------|----------------|-------------|--------------------------------------------------|-----------------------------|---------------|-----|
| Protty et al. 2017 [59] | Patients who had experienced their first ACS (n = 57,668) | Schizophrenia (n = 236) | ICD-10 diagnostic codes from electronic patient records of previous hospital admissions | LOS in days (M, CI, mean difference) | No | No | Patients with schizophrenia (M = 15.67 days, 95% CI 11.93–19.42 days) had a longer LOS compared to patients with no psychiatric diagnosis (M = 9.78 days, 95% CI 9.66–9.91), p < 0.001 | 7 Poor quality* |
| Sams et al. 2012 [60] | Patients undergoing total hip arthroplasty (n = 23,444) | Psychosis (n = NR) | Charlson and Elixhauser Comorbidity Indices | LOS in days (M, medians, no SD) | No | No | Patients with psychosis had longer elective LOS (M = 5.1 days) compared to the entire sample (M = 4.4 days) Patients with psychosis had longer nonelective LOS (M = 9.32 days) compared to the entire sample (M = 7.24 days), p < 0.001 | 7 Poor quality* |
| Sayers et al. 2007 [40] | Patients ≥65 years old with congestive heart failure (n = 21,429) | Bipolar disorder (n = 58) Psychosis (n = 534) | Identified using AHRQ's Clinical Classifications software (ICD-9 diagnostic codes) | Estimated additional mean LOS in days over 12-month study period | Elixhauser Comorbidity Index | Age, sex, race, SES | Patients with bipolar disorder had an estimated additional mean LOS of 1.43 days (32%, p = 0.02) and patients with psychosis had an estimated additional mean of 1.06 days (24%, p < 0.001) compared to patients with no psychiatric comorbidity | 8 Good quality |
| Schoepf et al. 2014 [41] | Patients admitted for medical treatment across three hospitals (n = 15,598) | Schizophrenia (n = 1,418) | ICD-10 diagnostic codes | LOS in days (M, SD) | No | No | Patients with schizophrenia (M = 8.1 days, SD = 0.6 days) had a longer average LOS at index hospitalisation compared to nonschizophrenic controls (M = 3.4 days, SD = 1.0 days), p < 0.001 | 7 Poor quality* |
| Siddiqui et al. 2018 [61] | Patients admitted with one of five chronic medical conditions: cancer (lung, colorectal), COPD, type 2 diabetes, ischaemic heart disease, or stroke (n = 16,898) | Schizophrenia (n = 34) | ICD-10 diagnostic codes entered as secondary diagnosis during the course of the hospital admission | LOS in days (M, SD) over the 5-year study period (obtained directly from author) | Charlson Comorbidity Index | Age, sex, SES, financial year, primary physical diagnosis | The average LOS was longer for patients with schizophrenia (M = 9.9 days, SD = 8.7 days) compared to patients without (M = 5.5 days, SD = 8.7 days)† Using fully adjusted negative binomial regression, schizophrenia was associated with a 91.2% longer LOS (95% CI 39.3%–162.6%)† | 8 Good quality |
| Sporinova et al. 2019 [42] | All adults (18+ years) in Alberta, Canada, with at least one of the following chronic diseases in 2012: asthma, congestive heart failure, myocardial infarction, diabetes, epilepsy, hypertension, chronic pulmonary disorder, and chronic kidney disease (n = 991,445) | Schizophrenia (n = 13,320) | ICD-10 diagnostic codes | Mean total LOS (days) for chronic disease admissions and mean total LOS for admissions relating to ACSCs, over the 3-year study period | No | No | Patients with schizophrenia had a longer LOS for chronic disease admissions (1.5, 1.3–1.7) compared to patients with no mental health conditions (0.6, 0.6–0.6). Patients with schizophrenia had a longer LOS for admissions relating to ACSCs (1.00, 0.80–1.10) compared to patients with no mental health conditions (0.46, 0.45–0.47) | 7 Poor quality |
| Tarrier et al. 2005 [62] | Burn injury patients admitted to an inpatient burns unit (n = 27) | Psychosis (n = 9) | ICD-10 diagnostic codes recorded in hospital's patient administration system | LOS in days (M, SD) | No | No | Patients with psychosis had a longer LOS (M = 40.4 days, SD = 37.1 days) compared to nonpsychotic controls (M = 13.0 days, SD = 18.9 days), p = 0.004 | 6 Poor quality* |

(*Continued*)

**Table 2.** (Continued)

| Authors | Population (n) | Comorbid SMI (n) | SMI assessment | Type of LOS | Control for comorbidities/ medical illness severity | Control for other variables | Main findings | NOS |
|---|---|---|---|---|---|---|---|---|
| Uldall et al. 1998 [63] | AIDS patients admitted to hospital for medical-surgical reasons (n = 1,295) | Bipolar disorder, schizophrenia, psychosis (n = NR) | ICD-9 diagnostic codes in hospital records | LOS in days (median) over the 3-year study period | No | No | The median LOS for patients with schizophrenia (22 days), psychosis (10 days), and bipolar disorder (9 days) was higher than the median stay for those with no psychiatric illness (7 days). However, this difference was not statistically significant | 7 Poor quality* |
| Vakharia et al. 2020 [64] | Patients on the Medicare Claims Database who had undergone primary total knee arthroplasty (n = 49,176) | Schizophrenia (n = 8,196) | ICD-9 diagnostic codes | In-hospital LOS (days) | No | No | Patients with schizophrenia had a significantly longer in-hospital LOS (3.73 days) compared to patients without (3.29 days) (p < .0001) | 7 Poor quality |
| Willers et al. 2018 [65] | Patients with ischaemic stroke (n = 46,350) | Schizophrenia, psychosis (n = 389) | The presence of at least one care event related to psychosis or schizophrenia according to ICD-10 diagnostic codes | Association between psychosis and increased LOS (beta) LOS in days (M, CI) over the 4-year study period | Diagnosis of atrial fibrillation and/or hypertension, ADL dependency, prior stroke, inpatient care prior to stroke, unconscious at arrival, NIHSS score at arrival | Age, sex, marital status, born outside EU, living alone, living arrangements | There was no association between psychosis and increased LOS in patients with ischaemic stroke (β = 0.06, 95% CI −0.06 to 0.19) (Comorbid psychosis LOS: M = 25.1 days, 95% CI 21.1–29.2 days, no psychosis LOS: M = 20.5 days, 95% CI 20.2–20.8 days) | 9 Good quality |

*If studies failed to adjust for physical comorbidities/illness severity in their analyses, they were deemed to be of poor quality, regardless of 'stars' recorded using the NOS.

**Independent *t* test carried out by author AR using data provided in paper.

†Raw data obtained through personal correspondence with authors.

Abbreviations: ACS, acute coronary syndrome; ACSC, ambulatory care sensitive condition; ADL, activities of daily living; AHRQ, Agency for Healthcare Research and Quality; AIDS, acquired immune deficiency syndrome; APR-DRG, All-Patient Refined Diagnostic Related Groups; CHD, coronary heart disease; CI, confidence interval; COPD, chronic obstructive pulmonary disorder; DSM-IV, Diagnostic and Statistical Manual of Mental Disorders-IV; DVT, deep vein thrombosis; EU, European Union; HCUP-NIS, Healthcare Cost and Utilization Project–National Inpatient Sample; HIV, human immunodeficiency virus; IBD, inflammatory bowel disease; ICD, International Statistical Classification of Diseases and Related Health Problems; IRR, incidence rate ratio; LOS, length of stay; M, mean; NIHSS, National Institute for Health Stroke Scale; NOS, Newcastle-Ottawa Scale; NR, not reported; OR, odds ratio; PAD, peripheral artery disease; PTSD, posttraumatic stress disorder; SD, standard deviation; SE, standard error; SES, socioeconomic status; SMI, severe mental illness; TBSA, total burn surface area; US, United States.

99%); therefore, the estimation of the overall pooled effect should be interpreted with caution. The pooled OR indicated that patients with SMI were significantly more likely to attend the emergency department than patients without (pooled OR = 1.97, 95% CI 1.41–2.76, p < 0.001; Fig 5). Because of the small number of analyses included in the meta-analysis, subgroup analysis to determine sources of heterogeneity was not appropriate [16]. All studies were of good quality, meaning this did not contribute to the variation between them. Examination of the funnel plot indicated that publication bias was possible (see S5 Appendix).

## The impact of comorbid SMI on use of primary care

Seven studies (10 separate analyses) looked at the impact of SMI on the use of primary care services (Table 5) [23,29,84,87–90]. Out of the 10 analyses, eight found that SMI was associated with increased primary care use. Five studies were of good quality and adjusted for physical illness severity, with the exception of that by Copeland and colleagues, who performed a cluster analysis [87], and Norgaard and colleagues, who provided unadjusted descriptive statistics [90]. One study found that there was no significant effect of SMI on primary care use in

| Study or Subgroup | SMI Mean | SMI SD | SMI Total | No SMI Mean | No SMI SD | No SMI Total | Weight | Std. Mean Difference IV, Random, 95% CI | Std. Mean Difference IV, Random, 95% CI |
|---|---|---|---|---|---|---|---|---|---|
| Attar et al. 2019 | 7.971 | 9.41 | 726 | 9.236 | 19.33 | 1452 | 6.1% | −0.08 [−0.16, 0.01] | |
| Chen et al. 2019 | 11.87 | 16.11 | 580 | 12.67 | 16.51 | 1740 | 6.1% | −0.05 [−0.14, 0.05] | |
| Banta et al. 2010 | 6.7 | 7.3 | 564 | 6.9 | 9.4 | 14933 | 6.1% | −0.02 [−0.11, 0.06] | |
| Sporinova et al. 2019 (ACSCs) | 1 | 8.83 | 13320 | 0.46 | 4.66 | 835149 | 6.1% | 0.11 [0.10, 0.13] | |
| Hsieh et al. 2012 | 1.7 | 1.3 | 4067 | 1.5 | 1.8 | 12201 | 6.1% | 0.12 [0.08, 0.15] | |
| Willers et al. 2018 | 25.1 | 40.7 | 389 | 20.5 | 32.8 | 45961 | 6.1% | 0.14 [0.04, 0.24] | |
| Falsgraf et al. 2017 (bipolar disorder) | 10.1 | 14.7 | 212 | 6.2 | 14.4 | 26290 | 6.0% | 0.27 [0.14, 0.41] | |
| Krein et al. 2006 | 12 | 15.9 | 18273 | 8.2 | 11.4 | 18273 | 6.1% | 0.27 [0.25, 0.30] | |
| Protty et al. 2017 | 15.7 | 29.4 | 236 | 9.8 | 15.3 | 57432 | 6.0% | 0.38 [0.26, 0.51] | |
| Menendez et al. 2013 | 11 | 21 | 64000 | 7.2 | 8.3 | 10605449 | 6.1% | 0.45 [0.44, 0.46] | |
| Siddiqui et al. 2018 | 9.9 | 8.7 | 34 | 5.5 | 8.7 | 16864 | 5.4% | 0.51 [0.17, 0.84] | |
| Gholson et al. 2018 | 3.9 | 2.14 | 953 | 3.2 | 1.3 | 504887 | 6.1% | 0.54 [0.47, 0.60] | |
| Falsgraf et al. 2017 (schizophrenia) | 14.2 | 21.7 | 344 | 6.2 | 14.4 | 26158 | 6.0% | 0.55 [0.44, 0.66] | |
| Hendrie et al. 2014 | 58.9 | 94.2 | 757 | 31.1 | 44.1 | 30831 | 6.1% | 0.61 [0.53, 0.68] | |
| Sporinova et al. 2019 (chronic conditions) | 1.5 | 11.78 | 13320 | 0.6 | 0.02 | 835149 | 6.1% | 0.61 [0.59, 0.63] | |
| Tarrier et al. 2005 | 40.4 | 37.1 | 9 | 13 | 18.9 | 18 | 3.4% | 1.02 [0.16, 1.87] | |
| Schoepf et al. 2014 | 8.1 | 0.6 | 1418 | 3.4 | 1 | 14180 | 6.1% | 4.84 [4.77, 4.92] | |
| **Total (95% CI)** | | | **119202** | | | **13046967** | **100.0%** | **0.59 [0.36, 0.83]** | |

Heterogeneity: Tau² = 0.24; Chi² = 15432.12, df = 16 (P < 0.00001); I² = 100%
Test for overall effect: Z = 4.93 (P < 0.00001)

**Fig 3. The impact of SMI on length of hospital stay (days).** Standardised mean difference values are presented. Two means were entered into the meta-analysis from Falsgraf and colleagues (2017) [54], as data were presented for patients with bipolar disorder and schizophrenia separately. Two means were entered for Sporinova and colleagues (2019) [42] as length of stay was looked at separately in patients hospitalised for chronic physical conditions and patients hospitalised for ACSCs. ACSC, ambulatory care sensitive condition; CI, confidence interval; SD, standard deviation; SMI, severe mental illness; Std., standard.

epilepsy patients [84]. Lichstein and colleagues reported that patients with schizophrenia were less likely to use medical homes for medical disorders (a model of primary care in the US) compared with patients without schizophrenia or depression [89]. Because of heterogeneity across outcomes, meta-analysis was not possible.

## Discussion

This systematic review and meta-analysis aimed to understand the impact of SMI on use of general inpatient, emergency, and primary care services. The evidence showed that SMI leads to increased use of general medical services. More specifically, having an SMI is associated with increased inpatient admissions, increased length of hospital stay, higher 30-day readmission rates, more emergency room attendances, and increased use of primary care services, for nonpsychiatric reasons. The results of this review highlight the extent to which patients with SMI need targeted and effective interventions and system-wide integrated mental and physical healthcare.

The majority of studies indicated that nonpsychiatric inpatient admissions, LOS, hospital readmission rates, and emergency department use were increased in medical patients with SMI compared to patients without SMI. This was confirmed with meta-analyses that showed that patients with SMI were more likely to have an inpatient admission, had hospital stays that were increased by 0.59 days, and were more likely to be readmitted within 30 days compared to patients without SMI. Meta-analysis also showed that patients with SMI were more likely to attend the emergency department. Most studies included in this review also found that SMI was associated with increased use of primary care services. These findings are in line with previous reviews that have shown that health service utilisation is increased in patients with psychiatric comorbidity [5–7]. However, this is, to our knowledge, the first time that the impact of SMI on the use of general medical services has been systematically reviewed and meta-analysed. We believe these findings highlight the need for system-wide integration of mental and physical health services, particularly in secondary care. Although it is possible that increases in the use of primary care services associated with SMI are reflecting the provision of integrated care already in place, we do not believe this is the case for specialist secondary care services.

**Table 3. The impact of comorbid SMI on hospital readmissions.**

| Authors | Population (n) | Comorbid SMI (n) | SMI assessment | Type of readmission | Control for comorbidities/medical illness severity | Control for other variables | Main findings | NOS |
|---|---|---|---|---|---|---|---|---|
| Ahmedani et al. 2015 [66] | Patients hospitalised for heart failure, acute myocardial infarction, or pneumonia (n = 89,406) | Bipolar disorder (n = 696) Schizophrenia (n = 257) Other psychoses (n = 502) | ICD-9 diagnostic codes for SMI recorded at least twice in any secondary care setting in the 12 months prior to hospitalisation | 30-day hospital readmission (all-cause) | No | No | 30-day readmission rates for bipolar disorder (21.2%, p < 0.001), schizophrenia (21.1%, p < 0.05), and other psychoses (30.4%, p < 0.001) were significantly higher than rates for patients with no psychiatric diagnosis (16.5%) | 7 Poor quality* |
| Ali et al. 2017 [67] | Patients admitted for total hip arthroplasty in the UK (Hospital Episode Statistics, NHS) (n = 514,455) | Psychosis (n = 960) | ICD-10 diagnostic codes | 30-day hospital readmission (all-cause, surgical, RTT) | 30+ comorbid conditions included in analysis, type of surgical procedure, RTT during index admission | Sex, age categories, SES, ethnicity, index LOS, number of prior emergency admissions | Psychosis was significantly associated with increased risk of all-cause 30-day readmission (OR = 1.51, 95% CI 1.23–1.86, p < 0.001), surgical 30-day readmission (OR = 1.73, 95% CI 1.33–2.25, p < 0.001), and RTT 30-day readmissions (OR = 1.83, 95% CI 1.16–2.87, p = 0.009) | 8 Good quality |
| Ali et al. 2019 [68] | Patients undergoing total knee arthroplasty in the UK (Hospital Episode Statistics, NHS) (n = 566,323) | Psychosis (n = 811) | ICD-10 diagnostic codes | 30-day all-cause, surgical, and RTT readmission rates | Diabetes, hypertension, arrythmia, valvular disease, congestive heart failure, PVD, chronic lung disease, lung circulation disorders, cancer, renal disease, dementia, alcohol and drug abuse, depression, other mental health disorder, liver disease, peptic ulcer, paraplegia, anaemia, coagulopathy, fluid and electrolyte disorders, hypothyroidism, other neurological diseases, rheumatic disorders, previous pneumonia, previous AMI, previous stroke | Age, gender, ethnic group, SES, type of knee replacement, year of surgery, number of emergency admissions in the previous year, RTT during index admissions, index LOS, primary diagnosis | Patients with psychosis had an increased risk of all-cause (OR = 1.69, 95% CI 1.37–2.08, p < 0.001), surgical (OR = 1.51, 95% CI 1.14–2.00, p = 0.0039) and RTT (OR = 2.52, 95% CI 1.49–4.24, p < 0.001) 30-day readmission compared to patients without psychosis | 9 Good quality |
| Chwastiak et al. 2014 [69] | Patients with diabetes who had a medical-surgical hospitalisation (n = 80,907) | Bipolar disorder, schizophrenia, psychotic disorders, delusional disorders, nonorganic psychoses (n = 1,820) | ICD-9 diagnostic codes | 30-day hospital readmission, readmission throughout study period (24 months) (for medical-surgical illness) | Elixhauser Comorbidity Index | Age, gender, payer type (US), number of hospitalisations in the 12 months prior, whether the index hospitalisation was through the ED, LOS | Having SMI was associated with an increased risk of 30-day hospital readmission (OR = 1.24, 95% CI 1.07–1.44, p = 0.006) Having SMI was also associated with a 14% greater risk of readmission throughout the study period (24 months) (HR = 1.14, 95% CI 1.05–1.23, p = 0.002) | 8 Good quality |
| Clark et al. 2013 [70] | Patients with alcohol withdrawal admitted to medical ICU** (n = 1,178) | Bipolar disorder (n = 121) Schizophrenia (n = 50) | ICD-9 diagnostic codes for SMI recorded twice in the preceding 3 years | Likelihood of having ≥3 hospital readmissions in the 12 months following discharge (all-cause) | Charlson Comorbidity Index; UHC severity of illness quartiles; diagnosis of depression and anxiety | Age, homelessness | Bipolar disorder was associated with a higher risk of having three or more readmissions to hospital in the year following initial discharge (OR = 1.93, 95% CI 1.08–3.46, p = 0.027). Schizophrenia was not (OR = 1.61, 95% CI 0.71–3.63, p = 0.252) | 7 Good quality |
| Dailey et al. 2013 [71] | Patients admitted for orthopaedic surgical procedures (n = 3,261) | Psychosis (n = 53) | Elixhauser Comorbidity Index | 30-day hospital readmission (all-cause) | No | No | Psychosis was associated (unadjusted analysis) with an increased risk of 30-day hospital readmission (OR = 3.58, 95% CI 1.59–8.08, p = 0.001) | 6 Poor quality* |
| Daratha et al. 2012 [72] | Patients admitted to hospital for medical illness (n = 925,705) | Bipolar disorder (n = 9,019) Schizophrenia (n = 6,868) | ICD-9 diagnostic codes recorded as secondary diagnoses on index admission to hospital | Likelihood of subsequent medical hospitalisations following index hospitalisation over the follow-up period (ranged from 1 to 72 months postsurgery) | Elixhauser Comorbidity Index, substance disorders, and interactions between substance disorders and SMI | Age, gender, index hospitalisation primary diagnosis, LOS, index hospitalisation via ED, primary payer, 12-month count of previous hospitalisations | Patients with bipolar disorder were at an increased risk of further inpatient admissions compared to those without during the study period (HR = 1.13, 99% CI 1.08–1.17, p < 0.001) However, patients with comorbid schizophrenia were no more likely than those without to return for inpatient medical (HR = 1.00, 99% CI 0.95–1.05, p = 0.92) | 7 Good quality |
| Davydow et al. 2016 [26] | Patients hospitalised for ACSCs (n = 5,945,540) | Bipolar disorder (n = 25,648) Schizophrenia (n = 42,558) | Diagnosis recorded in the Danish Civil Registration System | 30-day hospital readmission (for same ACSC or different ACSC) | Charlson Comorbidity Index | Age, sex, calendar period, marital status, education level, substance abuse, primary care use | Bipolar disorder and schizophrenia were examined together. SMI was associated with an increased risk of 30-day readmission for the same ACSC (IRR = 1.13, 95% CI 1.04–1.23, p < 0.001) and 30-day readmission for a different ACSC (IRR = 1.30, 95% CI 1.20–1.41, p < 0.001) | 9 Good quality |
| Feller et al. 2016 [73] | Patients admitted to general medical hospital with HIV (n = 16,558) | Psychosis (n = 1,872) | ICD-9 codes present on patient discharge | 30-day hospital readmission (medical-surgical) | Elixhauser Comorbidity Index; diagnosis of depression; substance abuse | Sex, SES, race, location, ever left hospital against medical advice, inpatient admissions, ED admissions, access to stable housing | Psychosis was associated with an increased risk of 30-day hospital readmission (OR = 1.43, 95% CI 1.27–1.62, p < 0.01) | 8 Good quality |

*(Continued)*

**Table 3.** (*Continued*)

| Authors | Population (n) | Comorbid SMI (n) | SMI assessment | Type of readmission | Control for comorbidities/medical illness severity | Control for other variables | Main findings | NOS |
|---|---|---|---|---|---|---|---|---|
| Fleming et al. 2019 [74] | Patients admitted for percutaneous cholecystostomy (n = 3,368) | Psychosis (n = 118) | ICD-9 diagnostic codes | 30-day hospital readmission (all-cause) | No | No | 2.72% (n = 19) of patients who were readmitted within 30 days were psychotic compared with 3.71% (n = 99) of patients who were not readmitted. There was no significant difference between these groups (p = 0.21) | 7 Poor quality* |
| Jorgensen et al. 2017 [75] | Patients admitted with heart failure (n = 36,718) | Schizophrenia (n = 108) | Identified from the Danish Schizophrenia Registry | 28-day hospital readmission (all nonpsychiatric) | Previous myocardial infarction, stroke, COPD, hypertension, diabetes; left ventricular ejection fraction (measure of heart failure severity) | Age, sex, alcohol intake, smoking habits | Patients with schizophrenia did not have a higher risk of 28-day hospital readmission (OR = 1.77, 95% CI 0.79–3.92) | 8 Good quality |
| Kheir et al. 2018 [76] | Patients admitted for total joint arthroplasty (n = 579) | Bipolar disorder, schizophrenia (n = 156) | ICD-9 diagnostic codes | 90-day, 12-month, and 24-month aseptic hospital readmission (related to index surgery) | Charlson Comorbidity Index | Age, gender, BMI, joint type, LOS, operative time, primary versus revision surgery | Patients with SMI were more likely to experience aseptic hospital readmission at 90 days (OR = 2.92, 95% CI 1.45–5.88, p = 0.003), 1 year (OR = 2.24, 95% CI 1.39–3.60, p = 0.001), and 2 years (OR = 1.83, 95% CI 1.18–2.86, p = 0.008) | 7 Good quality |
| Lau et al. 2017 [77] | Patients admitted to hospital with COPD (derivation cohort: n = 339,389; validation cohort: n = 258,113) | Psychosis (derivation cohort: n = 22,228; validation cohort: n = 13,745) | Extracted from the HCUP SID | 30-day hospital readmission (related to COPD) | Alcohol abuse, anaemia, congestive heart failure, depression, diabetes, drug abuse, liver disease, solid tumour without metastases | Age, gender, race, income, payer (US) | Derivation cohort: Psychosis was associated with an increased risk of 30-day hospital readmission (OR = 1.19, 95% CI 1.13–1.25, p < 0.01) Validation cohort: Psychosis was associated with an increased risk of 30-day hospital readmission (OR = 1.16, 95% CI 1.08–1.24, p < 0.01) | 6 Good quality |
| Lu et al. 2017 [78] | Patients admitted to hospital with decompensated heart failure (n = 611) | Bipolar disorder (n = 11) Schizophrenia (n = 40) | Electronic medical records—diagnostic codes not specified | 30-day hospital readmission related to heart failure; any hospital readmission over 3-year study period (all-cause) | Hypertension, diabetes, chronic kidney disease, coronary artery disease; heart failure severity was measured using peak troponin I, left ventricular ejection fraction, B-type natriuretic peptide | Age, sex, living situation, marital status | Patients with schizophrenia (OR = 4.92, 95% CI 2.49–9.71, p < 0.001) and patients with bipolar disorder (OR = 3.44, 95% CI 1.19–10.00, p = 0.02) had increased risk of 30-day hospital readmission Patients with schizophrenia (2.33, 95% CI 1.51–3.61, p < 0.001) and patients with bipolar disorder (OR = 2.08, 95% CI 1.05–4.11, p = 0.03) also had increased risk for any hospital readmission over the study period | 8 Good quality |
| Moore et al. 2017 [79] | All nonmaternal inpatients over a 1-year period in 18 US states (n = 10,777,210) | Psychosis (n = 505,575) | ICD-9 codes listed on discharge record | 30-day hospital readmission (all-cause) | Elixhauser Comorbidity Index | No | Psychosis was associated with increased risk of 30-day hospital readmission (OR = 1.34, 95% CI 1.33–1.35, no p-value reported) | 8 Good quality |
| Paxton et al. 2015 [80] | Patients admitted for total hip arthroplasty (n = 12,030) | Psychosis (n = 755) | Elixhauser Comorbidity Index | 30-day hospital readmission (related to index surgery) | Elixhauser Comorbidity Index; ASA physical status score; in-hospital medical complications; in-hospital surgical complications | Age, sex, race, BMI, discharge disposition, LOS, whether surgeon had total joint arthroplasty fellowship, surgeon's average yearly volume of procedures performed, hospital volume | Psychosis was associated with increased risk of 30-day hospital readmission (OR = 1.32, 95% CI 1.03–1.69, p = 0.028) | 8 Good quality |
| Prabhakaran et al. 2020 [81] | Patients >65 years with admissions for fall-related injuries on the Nationwide Readmissions Database developed for the HCUP (n = 358,581) | Psychosis (n = 15,295) | Elixhauser Comorbidity Index | Likelihood of a fall-related readmission within the 1-year study period | Elixhauser Comorbidity Index, illness severity, discharge disposition | Age, sex, insurance status, initial LOS, initial total hospital costs | Patients with psychosis had a significantly higher chance of a fall-related readmission compared to those without (OR = 1.16, 95% CI 1.09–1.23, p < .001) | 9 Good quality |
| Shah et al. 2018 [82] | Patients hospitalised with chronic pancreatitis (n = 25,259) | Psychosis (n = 1921) | Elixhauser Comorbidity Index | 30-day hospital readmission (all-cause) | Elixhauser Comorbidity Index; acute pancreatitis, pseudocyst, benign pancreatic neoplasms, cholangitis, pancreatic surgery, endoscopic retrograde cholangiopancreatography | Age, gender, household income, alcohol abuse, smoking, primary payer information, weekend versus weekday admission, LOS, discharge disposition, hospital ownership status, bed size, metropolitan status | Psychosis was associated with increased risk of 30-day hospital readmission (HR = 1.12, 95% CI 1.03–1.23, p = 0.007) | 8 Good quality |
| Singh et al. 2016 [83] | Older adults (66+) hospitalised for COPD (n = 135,498) | Psychosis (n = 4,511) | Elixhauser Comorbidity Index | 30-day hospital readmission (all-cause) | Use of mechanical ventilator, ICU admission during hospitalisation; comorbid psychological disorders | Age, sex, region, year of discharge, discharge destination, race, SES, LOS | Psychosis was associated with increased risk of 30-day hospital readmission (OR = 1.18, 95% CI 1.10–1.27) | 8 Good quality |

(*Continued*)

**Table 3.** (Continued)

| Authors | Population (n) | Comorbid SMI (n) | SMI assessment | Type of readmission | Control for comorbidities/medical illness severity | Control for other variables | Main findings | NOS |
|---------|---------------|------------------|----------------|--------------------|---------------------------------------------------|----------------------------|---------------|-----|
| Vakharia et al. 2020 [64] | Patients on the Medicare Claims Database who had undergone primary total knee arthroplasty (n = 49,176) | Schizophrenia (n = 8,196) | ICD-9 diagnostic codes | Likelihood of 90-day readmission after total knee arthroplasty | No | No | Patients with schizophrenia had a significantly higher incidence (18.26% versus 12.07%) and higher odds of 90-day readmission after surgery (OR = 1.58, 95% CI 1.48–1.69, p < .0001) compared to control patients without schizophrenia | 7 Poor quality* |

*If studies failed to adjust for physical comorbidities/illness severity in their analyses, they were deemed to be of poor quality regardless of 'stars' recorded using the NOS.

**Seventy-seven percent of index admissions were for medical illnesses other than alcohol withdrawal; therefore, this study was included in the review.

Abbreviations: ACSC, ambulatory care sensitive condition; AMI, xxxx; ASA, American Society of Anesthesiologists; BMI, body mass index; CI, confidence interval; COPD, chronic pulmonary obstructive disorder; ED, emergency department; HCUP SID, Healthcare Cost and Utilization Project Stat Inpatient Database; HIV, human immunodeficiency virus; HR, hazard ratio; ICD, International Statistical Classification of Diseases and Related Health Problems; ICU, intensive care unit; IRR, incidence rate ratio; LOS, length of stay; NHS, National Health Service; NOS, Newcastle-Ottawa Scale; OR, odds ratio; PVD, peripheral vascular disease; RTT, return to theatre; SES, socioeconomic status; SMI, severe mental illness; UHC, University Health System Consortium; UK, United Kingdom; US, United States.

Subgroup analyses revealed some interesting findings surrounding factors that might impact upon how SMI affects health service utilisation. The likelihood of inpatient admission differed across SMI subtypes, with schizophrenia patients having the highest risk of admission and patients with bipolar disorder having the lowest. This mirrored results relating to LOS, which was longest in those with schizophrenia and shortest in those with bipolar disorder. Results relating to LOS should be interpreted with caution, however. Subgroup analysis revealed that only studies of poor quality showed that LOS differed significantly between those with and without SMI, which suggests that factors adjusted for in good-quality studies such as physical comorbidities and illness severity might better explain hospital LOS. Interestingly, the likelihood of 30-day readmission amongst patients with SMI was substantially reduced in

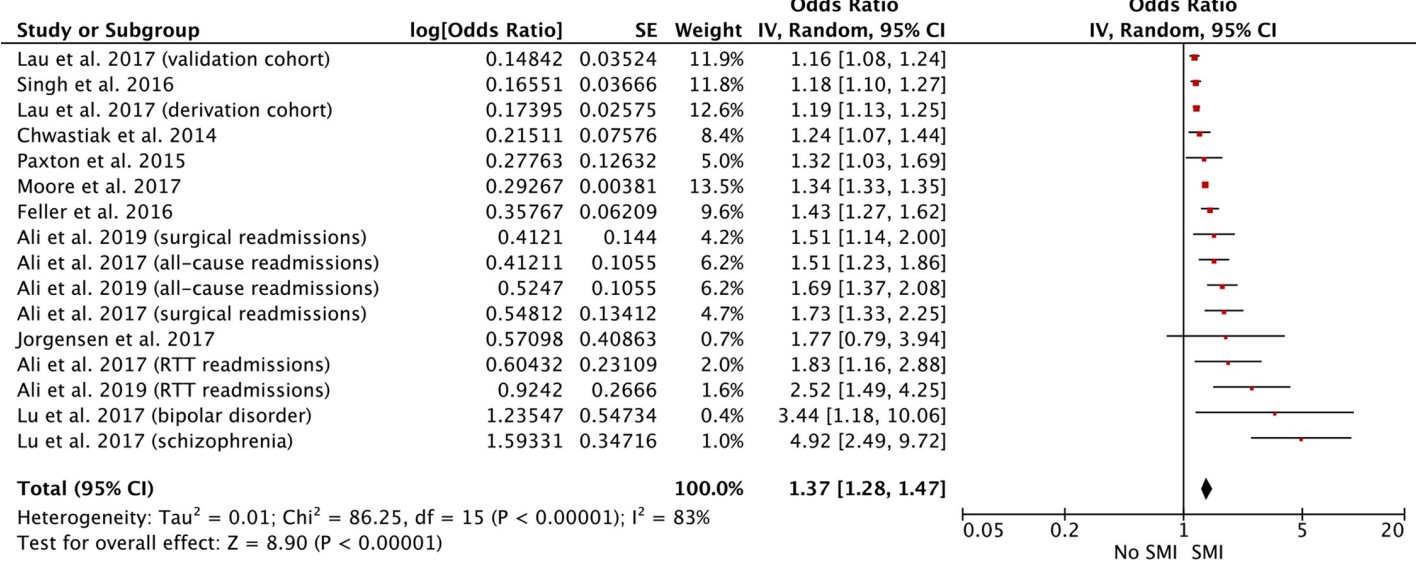

**Fig 4. The impact of SMI on 30-day hospital readmission rate.** Several studies included analyses for different samples and included different outcomes. Therefore, several effect estimates have been entered from the same studies. CI, confidence interval; RTT, return to theatre; SE, standard error; SMI, severe mental illness.

**Table 4. The impact of comorbid SMI on ED use.**

| Authors | Population (n) | Comorbid SMI (n) | SMI assessment | Type of ED use | Control for comorbidities/medical illness severity | Control for other variables | Main findings | NOS |
|---|---|---|---|---|---|---|---|---|
| Bresee et al. 2012 [23] | All patients on the Administrative Database of Alberta Health and Wellness (1995–2006) (n = 2,310,391) | Schizophrenia (n = 28,755) | ICD-9 and ICD-10 diagnostic codes from physician claims data and hospital discharge data | ED visit over the 2-year study period; ≥1 yearly ED visit | Diagnosis of coronary artery disease and/or diabetes | Age, sex, SES, urban versus rural dwelling | Patients with schizophrenia were more likely to visit the ED over the 2-year study period (OR = 3.57, 95% CI 3.44–3.71) and were also more likely to have ≥1 ED visit per year (OR = 3.47, 95% CI 3.38–3.56) | 8 Good quality |
| Clark et al. 2013 [70] | Patients with alcohol withdrawal admitted to medical ICU** who did not die and were not rehospitalised in the 6-year study period (n = 656) | Bipolar disorder (n = 121) Schizophrenia (n = 50) | ICD-9 diagnostic codes for SMI recorded twice in the preceding 3 years | Likelihood of ED or urgent care visit within a year of hospital discharge | Charlson/Deyo score, illness severity score based on admission diagnosis and comorbidities, depression, anxiety | Age, gender, race, payer source, smoker, homelessness | Having bipolar disorder was associated with a higher risk of an ED or urgent care visit within 1 year of hospital discharge (HR = 2.03, 95% CI 1.24–3.62, p = 0.008), but schizophrenia was not (HR = 1.97, 95% CI 0.72–5.31, p = 0.124) | 7 Good quality |
| Hsieh et al. 2012 [20] | Patients on the Taiwanese National Health Research Institute (case-control design) (n = 16,268) | Bipolar disorder (n = 4,067) | Acute admission ICD-9 diagnostic codes | Frequency of nonpsychiatric ED visits for those with bipolar disorder and matched controls over 2-year study period | No | Age, gender, urbanisation level of the residential area, monthly income | Patients with bipolar disorder had a significantly higher number of emergency care visits (M = 2.50, SD = 3.35) compared to matched controls (M = 1.51, SD = 1.52) over the study period (p < 0.001) | 6 Poor quality* |
| Hunter et al. 2015 [29] | Costliest 5% of Veterans Association patients (n = 261,515) | Bipolar disorder, schizophrenia, other psychosis (n = 33,119) | ICD-9 diagnostic codes and chronic condition indicators established by AHRQ | Number of ED visits over the 12-month study period | AHRQ comorbidity measures | Age, sex, race/ethnicity, marital status, documented homelessness during year of investigation, correlation within facilities | Patients with SMI had more ED visits over the study period (M = 2.6) compared to patients with no mental health conditions (M = 1.8) (p < 0.001) | 8 Good quality |
| Kheir et al. 2018 [76] | Patients admitted for total joint arthroplasty (n = 579) | Bipolar disorder, schizophrenia (n = 156) | ICD-9 diagnostic codes | Preoperative ED visit | Cohort and controls matched using Charlson Comorbidity Index | Also matched on age, gender, BMI, joint, LOS, operative time, revision | A significantly higher proportion of patients with SMI had a preoperative ED visit (19/156, 12%) compared to matched controls (15/423, 3.5%) (p < 0.001) | 7 Good quality |
| Kurdyak et al. 2017 [31] | Patients with diabetes (n = 1,131,375) | Schizophrenia (n = 26,259) | Ontario Health Insurance Plan records detailing three schizophrenia-related physician visits in 36 months, or a hospitalisation for schizophrenia | Number of ED visits for diabetic complications; number of ED visits for any nonmental health reason excluding trauma over the 2-year study period | Johns Hopkins ACG System, duration of diabetes | Age, sex, rural residence, neighbourhood income, neighbourhood material deprivation, past year service use | Patients with schizophrenia had an increased risk of an ED visit for a diabetic complication (OR = 1.34, 95% CI 1.28–1.41) and an increased risk of an ED visit for any nonmental health reason excluding trauma (OR = 1.72, 95% CI 1.68–1.77) | 8 Good quality |
| Lafeuille et al. 2014 [32] | Patients with substance dependence/abuse, obesity, diabetes, metabolic syndrome, hyperlipidaemia, hypertension, coronary artery disease, congestive heart failure, HIV, hepatitis C, or COPD (n = 49,304) | Schizophrenia (n = 24,652) | ICD-9 diagnostic codes—at least two primary or secondary schizophrenia diagnoses recorded during study period | Number of ED visits over the 10-year study period | Patients were matched 1:1 based on propensity scores using the Charlson Comorbidity Index | Patients were matched 1:1 based on propensity scores on age, gender, state, and year of index admission | Compared to matched nonschizophrenic controls, patients with schizophrenia had significantly higher rates of ED visits over the study period (IRR = 2.08, 95% CI 1.95–2.23, p < 0.005) | 8 Good quality |
| Minen et al. 2014 [39] | Patients who visited the ED with a primary diagnosis of migraine (n = 2,872) | Bipolar disorder (n = NR) | ICD-9 diagnostic codes recorded in the Partners Research Patient Data Registry, Massachusetts | Number of ED visits over the 10-year study period | No | No | Patients with bipolar disorder had 1.85 times more ED visits than patients with other psychiatric disorders. No inferential analyses were carried out on individual psychiatric conditions, and no comparison was made between patients with bipolar disorder and those without any psychiatric diagnosis | 7 Poor quality* |
| Norbeck et al. 2019 [34] | Homeless male veterans in urban and rural settings in the US (n = 156) | Bipolar disorder (n = 39) | Self-reported bipolar disorder | Self-reported use of the emergency room over the past 3 months | No | No | Having bipolar disorder was not associated with an increase in the use of the emergency room for medical reasons (no p-value reported) | 4 Poor quality* |
| Pugh et al. 2008 [84] | Patients (veterans) with epilepsy (n = 23,752) | Bipolar disorder, schizophrenia, other psychoses (n = 1,412) | ICD-9 diagnostic codes | Number of ED attendances over the 12-month study period | Epilepsy chronicity, epilepsy severity, physical comorbidities | Age, sex, race, marital status, patients with service-connected disability (therefore, no co-payment required) versus patients with a required co-payment | Epilepsy patients with SMI were more likely to attend an ED over the study period compared to epilepsy patients without psychiatric disease (OR = 1.4, 95% CI 1.2–1.7, p < 0.01) | 8 Good quality |
| Shim et al. 2014 [85] | Patients with diabetes (n = 340,786) | Schizophrenia (n = 23,913) | ICD-9 diagnostic codes from claims data | Total ED visits, ED visits related to diabetes, ED visits related to other medical causes over the 2-year study period | No | No | Diabetes patients with schizophrenia had more total ED visits per year (M = 7.5, SD = 8.1) compared to diabetic patients without schizophrenia (M = 4.7, SD = 4.4), p < 0.01 Diabetes patients with schizophrenia had more ED visits related to diabetes per year (M = 0.3, SD = 0.7) compared to diabetes patients without schizophrenia (M = 0.2, SD = 0.5), p < 0.01 Diabetes patients with schizophrenia had more ED visits related to other medical causes per year (M = 6.9, SD = 7.6) compared to diabetes patients without schizophrenia (M = 4.4, SD = 4.1), p < 0.01 | 7 Poor quality* |

(*Continued*)

**Table 4.** (Continued)

| Authors | Population (n) | Comorbid SMI (n) | SMI assessment | Type of ED use | Control for comorbidities/medical illness severity | Control for other variables | Main findings | NOS |
|---|---|---|---|---|---|---|---|---|
| Sporinova et al. 2019 [42] | All adults (18+ years) in Alberta, Canada, with at least one of the following chronic diseases in 2012: asthma, congestive heart failure, myocardial infarction, diabetes, epilepsy, hypertension, chronic pulmonary disorder, and chronic kidney disease (n = 991,445) | Schizophrenia (n = 13, 320) | ICD-10 diagnostic codes | ED visits for chronic disease per 1,000 patient-days over the 3-year study period | No | No | Patients with schizophrenia had more ED visits for chronic diseases (0.28, 0.24–0.31) than patients with no mental health conditions (n = 0.13, 0.13–0.14) | 7 Poor quality* |
| Wallace et al. 2019 [44] | Medical claims data from the HealthCore Integrated Research Database (US) (n = 33,660) | Schizophrenia (n = 6,732) | ICD-9 and ICD-10 diagnostic codes | Rates of all-cause ED visits in the year preceding schizophrenia diagnosis | No | Patients matched 1:4 on age, sex, and region of residence | Patients with schizophrenia had 3-fold more all-cause ED visits (32.5%) compared to their matched comparators without schizophrenia (11.9%) | 7 Poor quality* |
| Weilburg et al. 2018 [86] | Patients in the Medicare CMHCB-DP at Massachusetts General Hospital (n = 3,620) | Psychosis (n = 427) | ICD-9 diagnostic codes | Likelihood of ED use in the study period (5 years and 5 months) | HCC score | Age, sex, age at enrolment, poststudy survival, year of enrolment | Patients with psychosis were more likely to use the ED than those with no behavioural health conditions (OR = 1.42, 95% CI 1.14–1.77) | 8 Good quality |
| Wetmore et al. 2019 [45] | Patients on Medicare Claims Database who have Parkinson's disease (n = 52,103) | Psychosis (n = 2,778) | ICD-9 diagnostic codes | ED visits over the 6-year study period | Patients matched 1:4 based on number of comorbid conditions | Patients matched 1:4 on age, sex, race, index year of psychosis diagnosis | Patients with psychosis had more ED visits (1.5) than patients without psychosis (0.7) in the sixth year of follow-up | 8 Good quality |

*If studies failed to adjust for physical comorbidities/illness severity in their analyses, they were deemed to be of poor quality regardless of 'stars' recorded using the NOS.

ACG, Adjusted Clinical Groups; AHRQ, Agency for Healthcare Research and Quality; BMI, body mass index; CI, confidence interval; CMHCB-DP, Case Management for High-Cost Beneficiaries Demonstration Project; COPD, chronic obstructive pulmonary disorder; ED, emergency department; HCC, hierarchical condition category; HIV, human immunodeficiency virus; HR, hazard ratio; ICD, International Statistical Classification of Diseases and Related Health Problems; ICU, intensive care unit; IRR, incidence rate ratio; LOS, length of hospital stay; M, mean; NOS, Newcastle-Ottawa Scale; NR, not reported; OR, odds ratio; SES, socioeconomic status; SD, standard deviation; SMI, severe mental illness; US, United States.

studies carried out in the US. This perhaps reflects higher healthcare costs in the US [91], making patients less likely to seek readmission.

There are several factors that might explain why patients with SMI are using nonpsychiatric healthcare services more than those without. The primary reason for increased service use is likely the considerable rates of physical illness seen in patients with SMI. Significant increases in levels of obesity, metabolic syndrome, diabetes, cardiovascular disease, viral disease, respiratory tract disease, and musculoskeletal disease are seen alongside SMI, and illness severity is usually more pronounced in these patients [9]. Increased morbidity in SMI patients is largely down to a higher prevalence of modifiable risk factors [92], such as smoking [93], obesity [94], and alcohol and substance misuse [95]. Additionally, SMI is associated with physiological changes known to impact upon physical health. For example, changes in cortisol secretion associated with dysregulation of the hypothalamic pituitary adrenal (HPA) axis have been seen in patients with SMI [96,97]. Moreover, patients with SMI have increased levels of blood cytokines and circulating immune cell abnormalities [98,99], even at the early stages of mental illness [100]. Use of psychotropic medications has also been associated with increased obesity, dyslipidaemia, type 2 diabetes, and subsequent increased cardiovascular risk [101,102].

There is also evidence that disparities exist in the provision of healthcare for people with SMI, affecting the incidence and severity of physical disease. It is well documented that physical conditions in patients with SMI are underdiagnosed and suboptimally treated. SMI patients tend to have lower rates of medical and surgical intervention (e.g., cardiovascular stenting), and the quality of medical care, once received, can be substandard (e.g., levels of diabetes care) [10]. Moreover, the uptake of preventive strategies, such as cancer screening, is lower amongst those with SMI [103]. There are several reasons for these inequalities in health provision. Firstly, psychiatric symptoms may prevent the patient from seeking adequate physical healthcare. For example, cognitive impairment is often associated with SMI [104] and might impact

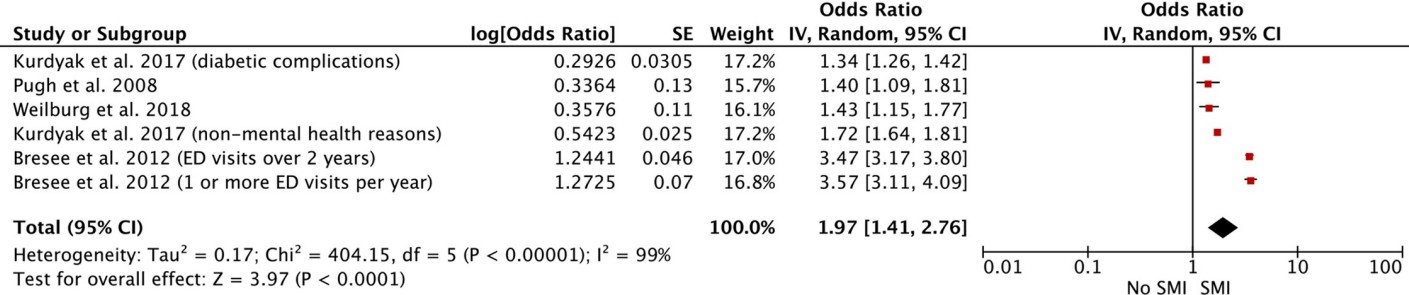

| Study or Subgroup | log[Odds Ratio] | SE | Weight | Odds Ratio IV, Random, 95% CI |
|---|---|---|---|---|
| Kurdyak et al. 2017 (diabetic complications) | 0.2926 | 0.0305 | 17.2% | 1.34 [1.26, 1.42] |
| Pugh et al. 2008 | 0.3364 | 0.13 | 15.7% | 1.40 [1.09, 1.81] |
| Weilburg et al. 2018 | 0.3576 | 0.11 | 16.1% | 1.43 [1.15, 1.77] |
| Kurdyak et al. 2017 (non–mental health reasons) | 0.5423 | 0.025 | 17.2% | 1.72 [1.64, 1.81] |
| Bresee et al. 2012 (ED visits over 2 years) | 1.2441 | 0.046 | 17.0% | 3.47 [3.17, 3.80] |
| Bresee et al. 2012 (1 or more ED visits per year) | 1.2725 | 0.07 | 16.8% | 3.57 [3.11, 4.09] |
| **Total (95% CI)** | | | **100.0%** | **1.97 [1.41, 2.76]** |

Heterogeneity: Tau² = 0.17; Chi² = 404.15, df = 5 (P < 0.00001); I² = 99%
Test for overall effect: Z = 3.97 (P < 0.0001)

**Fig 5. The impact of SMI on use of EDs.** Two studies performed analyses on different outcomes. Therefore, two effect estimates have been entered for these studies. CI, confidence interval; ED, emergency department; SE, standard error; SMI, severe mental illness.

upon a patients ability to access and understand health information and health services. Factors such as lack of motivation and self-neglect also often accompany SMI and will likely affect the extent to which a patient accesses medical services and adheres to medical advice [10]. Secondly, the stigma of SMI pervades all aspects of society, including healthcare. Stigmatisation of patients by physicians and other healthcare professionals can lead to diagnostic overshadowing and a lack of adequate care [105]. Diagnostic overshadowing might also be a result of a lack of training in physical or mental health or a lack of knowledge surrounding symptom recognition [106]. Moreover, it is possible that primary care physicians and psychiatrists feel unable to provide both physical and mental healthcare to SMI patients because of time constraints or lack of clinical training [10]. Thirdly, there is a known inverse association between socioeconomic status and mental illness [107]. Socioeconomic disadvantage is associated with poor access to healthcare. This association is likely to be stronger in some countries more than others, which is reflected in the results of the current study which showed that 30-day readmission is less likely for SMI patients in the US, where healthcare is costly.

All of these factors taken together increase the risk of treatment delay and the development of complications in patients with SMI. A lack of adequate physical healthcare for patients with SMI means that their physical symptoms are likely to be much worse when they finally present in general medical services, potentially leading to an increase in hospital admissions, rates of

**Table 5. The impact of comorbid SMI on primary care use.**

| Authors | Population (n) | Comorbid SMI (n) | SMI assessment | Type of primary care service use | Control for comorbidities/ medical illness severity | Control for other variables | Main findings | NOS |
|---|---|---|---|---|---|---|---|---|
| Bresee et al. 2012 [23] | All patients on the Administrative Database of Alberta Health and Wellness (1995–2006) (n = 2,310,391) | Schizophrenia (n = 28,755) | ICD-9 and ICD-10 diagnostic codes from physician claims data and hospital discharge data | Number of GP encounters in the 2-year study period, ≥4 yearly GP encounters | Diagnosis of coronary artery disease and/or diabetes | Age, sex, SES, urban versus rural dwelling | Patients with schizophrenia were more likely to have an encounter with their GP over the study period (OR = 7.57, 95% CI 5.92–9.69) and were more likely to have four or more GP encounters per year (OR = 3.60, 95% CI 3.49–3.71) | 8 Good quality |

*(Continued)*

**Table 5.** (Continued)

| Authors | Population (*n*) | Comorbid SMI (*n*) | SMI assessment | Type of primary care service use | Control for comorbidities/ medical illness severity | Control for other variables | Main findings | NOS |
|---------|-----------------|---------------------|----------------|----------------------------------|-----------------------------------------------------|------------------------------|---------------|-----|
| Copeland et al. 2009 [87] | Patients with diabetes (*n* = 201,357) | Schizophrenia (*n* = 13,025) | ICD-9 diagnostic codes from at least one Veteran's Association inpatient stay or two outpatient visits on different dates | Number of primary care visits over the 5-year study period | No | No | Cluster analysis identified four clusters of primary care use over the study period: (1) increasing use; (2) consistent use; (3) low to decreasing use; (4) high to decreasing use. The proportion of patients with diabetes only and diabetes + schizophrenia is presented below Increasing: diabetes only 7%, schizophrenia + diabetes 8% Consistent: diabetes only 33%, schizophrenia + diabetes 28% Low decreasing: diabetes only 52%, schizophrenia + diabetes 54% High decreasing: diabetes only 8%, schizophrenia + diabetes 10% The diabetes-only group dominates in the 'consistent use' cluster. However, patients with comorbid schizophrenia dominate in the other three clusters | 5 Poor quality |
| Hunter et al. 2015 [29] | Costliest 5% of Veterans Association patients (*n* = 261,515) | Bipolar disorder, schizophrenia, other psychosis (*n* = 33,119) | ICD-9 diagnostic codes and chronic condition indicators established by AHRQ | Number of primary care appointments over the 12-month study period | AHRQ comorbidity measures | Age, sex, race/ ethnicity, marital status, documented homelessness during year of investigation, correlation within facilities | Patients with SMI had more primary care visits over the study period (M = 6.1) compared to patients with no mental health conditions (M = 5.1) (*p* < 0.001) | 8 Good quality |

*(Continued)*

**Table 5.** (*Continued*)

| Authors | Population (*n*) | Comorbid SMI (*n*) | SMI assessment | Type of primary care service use | Control for comorbidities/ medical illness severity | Control for other variables | Main findings | NOS |
|---|---|---|---|---|---|---|---|---|
| Kontopantelis et al. 2015 [88] | Patients in UK registered with primary care practice (Different data are reported for each of the 12 fiscal years separately. We report sample sizes from the most recent year—2011/ 2012) (*n* = 5,069,748) | Schizophrenia, affective psychoses (bipolar disorder or other unspecified affective psychosis), other types of psychosis (*n* = 31,807) | Primary care Read Codes | Number of primary care visits over the 12-year study period. This includes face-to-face, telephone, and other (mail/ email, referrals, secondary care episode, other administrative tasks) | Hypertension, asthma, hypothyroidism, osteoarthritis, chronic kidney disease, coronary heart disease, epilepsy, COPD, cancer, stroke, heart failure, rheumatoid arthritis, dementia, and psoriasis | Patients matched on age, sex, and primary care practice | Patients with SMI consulted more in primary care than patients without SMI across the 12-year study period. The difference between these groups increased after the introduction of QOF in 2004 Consultation rates for people with SMI increased across all types over time. For people with SMI, the mean number of consultations was 92% higher in 2011/2012 compared with 2000/2001 (IRR = 1.92, 95% CI 1.91–1.93). For matched control cases, the increase was smaller at 75% (IRR = 1.75, 95% CI 1.74–1.75) | 8 Good quality |
| Lichstein et al. 2014 [89] | Patients with two or more chronic conditions: major depressive disorder, schizophrenia, hypertension, diabetes, hyperlipidaemia, seizure disorder, asthma, COPD (*n* = 105,542) | Schizophrenia (*n* = 10,166) | ICD-9 diagnostic codes | Use of medical homes over the 3-year study period. Medical homes provide a model of primary care that is patient-centred, comprehensive, team-based, coordinated, accessible, and focused on quality and safety | Total number of chronic conditions (between 2 and 8), included all diagnosis indicators for major depressive disorder, diabetes, asthma, hypertension, hyperlipidaemia, seizure disorder, and COPD | Age, gender, race, ethnicity | Patients with schizophrenia had an 8.2% lower probability of having a medical home visit over the study period compared to patients without schizophrenia or depression (*p* < 0.01) Patients with schizophrenia had one fewer (−1.02, SE = 0.10) medical home visits over the study period compared to patients without schizophrenia or depression (*p* < 0.01) | 8 Good quality |

(*Continued*)

**Table 5.** (Continued)

| Authors | Population (*n*) | Comorbid SMI (*n*) | SMI assessment | Type of primary care service use | Control for comorbidities/ medical illness severity | Control for other variables | Main findings | NOS |
|---|---|---|---|---|---|---|---|---|
| Norgaard et al. 2019 [90] | A matched study cohort selected from the Danish National Registers (*n* = 456,897) | Schizophrenia (*n* = 21,757) | ICD-8 and ICD-10 diagnostic codes | Number of general practice consultations 1 and 5 years after the index date | No | Age, sex, calendar time, cohabitation | Patients with schizophrenia had 82% (95% CI 78%–87%) more consultations than patients without schizophrenia after 1 year and 76% (95% CI 71%–80%) more after 5 years (6.05 versus 3.55 annual consultations) | 7 Poor quality* |
| Pugh et al. 2008 [84] | Patients (veterans) with epilepsy (*n* = 23,752) | Bipolar disorder, schizophrenia, other psychoses (*n* = 1,412) | ICD-9 diagnostic codes | High primary care use (top 20% of patients with high utilisation) over the 12-month study period | Epilepsy chronicity, epilepsy severity, physical comorbidities | Age, sex, race, marital status, patients with service-connected disability (therefore, no co-payment required) versus patients with a required co-payment | Epilepsy patients with SMI were no more likely to be frequent users of primary care over the study period than were epilepsy patients without psychiatric disease (OR = 0.9, 95% CI 0.7–1.1) | 8 Good quality |

*If studies failed to adjust for physical comorbidities/illness severity in their analyses, they were deemed to be of poor quality regardless of 'stars' recorded using the NOS.

Abbreviations: AHRQ, Agency for Healthcare Research and Quality; CI, confidence interval; COPD, chronic obstructive pulmonary disorder; GP, general practitioner; ICD, International Statistical Classification of Diseases and Related Health Problems; IRR, incidence rate ratio; M, mean; NOS, Newcastle-Ottawa Scale; OR, odds ratio; QOF, Quality Outcomes Framework; SE, standard error; SES, socioeconomic status; SMI, severe mental illness; UK, United Kingdom.

readmission, emergency department attendance, and use of primary care services, which we have described in this review.

## Limitations

The majority of studies included in this review were of good quality with large sample sizes. In most cases, SMI was defined using established diagnostic codes, and health service utilisation outcomes were obtained using medical record linkage. However, some studies (35%) were rated as poor quality, largely owing to failure to control for physical illness severity or the presence of comorbidities, meaning the strength of evidence differed across studies. Nevertheless, most studies showed that SMI led to an increase in the use of general medical services, and study quality was not a significant source of heterogeneity in all but one meta-analysis. However, most studies included in the meta-analysis looking at LOS were of poor quality, which should be taken into account when interpreting the result. Only peer-reviewed publications were included in the review, meaning that publication bias was likely. Despite carrying out literature searches on several databases using comprehensive search strategies, it is possible that retrieval of all relevant research was not complete.

$I^2$ values indicated considerable heterogeneity across studies. This was to be expected in a review of this kind, in which there is large variation between studies in terms of patient

population, hospital setting, and health system. Examining sources of variation revealed that certain outcomes differed according to SMI subtype and geographical location. Other patient and clinical characteristics likely explain the bulk of heterogeneity. This means that there is uncertainty around the magnitude of the impact of SMI on nonpsychiatric health service utilisation, and results of this review should be interpreted with this in mind.

In terms of the quality assessment of studies included in the review, the NOS has been criticised in terms of its subjectivity, and interrater reliability between authors and reviewers of studies has been found to differ significantly in that reviewers tend to view studies more favourably than the authors [108]. However, there is no gold standard when it comes to quality-assessment tools for observational studies [109]. Moreover, in the current review, we adopted quite stringent criteria for deciding when a study was poor quality (i.e., when a study did not adjust for severity of physical illness and/or the presence of physical comorbidities), meaning that the purported favourable view taken by reviewers might have been offset by that.

Although the focus of this review was on nonpsychiatric health service utilisation, it was not always possible to determine whether the health service outcome definitively excluded mental health treatment (e.g., use of the emergency department). Because of the nature of primary care, it was not possible to rule out the use of primary care services for psychiatric reasons in any of the studies included in the review. However, a recent study reported that nine out of 10 of the most common patient-reported reasons for primary care visits were nonpsychiatric [110], indicating that this outcome was likely to be mainly nonpsychiatric. It is very common for patients to have other psychiatric illnesses alongside SMI [111]. Unfortunately, because of the nature of the studies included, it was not within the scope of the current review to consider the impact of the overlap between SMI and other psychiatric conditions on nonpsychiatric health service utilisation. The studies included in this systematic review all described the impact of SMI on the use of general medical services in high-income countries, which affects the generalisability of the results. Future research should try to understand the impact of SMI, and more generally psychiatric comorbidity, on nonpsychiatric health service utilisation in low- and middle-income countries where health systems are not as well developed.

## Conclusions

The results of this systematic review and meta-analysis highlight the extent to which SMI impacts upon general medical services. Patients with SMI were more likely to have an inpatient admission, had hospital stays that were increased by 0.59 days, were more likely to be readmitted within 30 days, and were more likely to attend the emergency department compared to patients without SMI. Most studies included in this review also showed that SMI was associated with increased use of primary care services. Illustrating and quantifying this helps to build a case for system-wide integration of mental and physical healthcare. Additionally, the results of the meta-analyses might be used to guide clinicians, policy makers, and commissioners in the improvement of the delivery of physical healthcare for SMI patients. The Five Year Forward View for Mental Health for National Health Service England aims to improve early detection of physical illness in SMI patients through the implementation of physical healthcare screening, assessment, and intervention [112]. This is to be delivered across both primary and secondary care. Prevention, early detection, and timely delivery of treatment for physical illness will hopefully improve the physical health of patients with SMI and reduce use of nonpsychiatric healthcare services.

## Supporting information

**S1 Appendix. PRISMA checklist for the systematic review and meta-analyses.** PRISMA, Preferred Reporting Items for Systematic Reviews and Meta-Analysis.
(DOC)

**S2 Appendix. Literature search strategy.**
(DOCX)

**S3 Appendix. List of excluded studies.**
(DOCX)

**S4 Appendix. Subgroup analyses tables for meta-analyses.**
(DOCX)

**S5 Appendix. Funnel plots for meta-analyses.**
(DOCX)

## Author Contributions

**Conceptualization:** Amy Ronaldson.

**Data curation:** Amy Ronaldson.

**Formal analysis:** Amy Ronaldson, Lotte Elton, Simone Jayakumar, Anna Jieman.

**Methodology:** Amy Ronaldson, Lotte Elton, Kristoffer Halvorsrud.

**Project administration:** Kamaldeep Bhui.

**Resources:** Kamaldeep Bhui.

**Supervision:** Kamaldeep Bhui.

**Writing – original draft:** Amy Ronaldson.

**Writing – review & editing:** Lotte Elton, Simone Jayakumar, Anna Jieman, Kristoffer Halvorsrud, Kamaldeep Bhui.

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
