## [Editor Report · Decision Letter 0]

3 Feb 2020

Dear Dr Ronaldson, 

Thank you for submitting your manuscript entitled "The impact of severe mental illness on health service utilisation for non-psychiatric medical disorders: A systematic review and meta-analysis" for consideration by PLOS Medicine.

Your manuscript has now been evaluated by the PLOS Medicine editorial staff and I am writing to let you know that we would like to send your submission out for external peer review.

Kind regards,

Helen Howard, for Clare Stone PhD 

Acting Editor-in-Chief

PLOS Medicine 

plosmedicine.org

---

## [Decision Letter · Decision Letter 1]

10 Mar 2020

Dear Dr. Ronaldson,

Thank you very much for submitting your manuscript "The impact of severe mental illness on health service utilisation for non-psychiatric medical disorders: A systematic review and meta-analysis" (PMEDICINE-D-20-00254R1) for consideration at PLOS Medicine. 

Your paper was discussed among the editorial team and sent to independent reviewers, including a statistical reviewer. The reviews are appended at the bottom of this email and any accompanying reviewer attachments can be seen via the link below:

[LINK]

In light of these reviews, we will not be able to accept the manuscript for publication in the journal in its current form, but we would like to invite you to submit a revised version that fully addresses the reviewers' and editors' comments. You will appreciate that we cannot make a decision about publication until we have seen the revised manuscript and your response, and we expect to seek re-review by one or more of the reviewers. 

We hope to receive your revised manuscript by Mar 31 2020 11:59PM. Please email us (plosmedicine@plos.org) if you have any questions or concerns.

Please let me know if you have any questions. Otherwise, we look forward to receiving your revised manuscript in due course. 

Sincerely,

Richard Turner PhD, for Caitlin Moyer, Ph.D.

Associate Editor, PLOS Medicine

rturner@plos.org

We ask you to update your literature search to the end of 2019, say. 

Please begin the title "Severe mental illness and health-service ...". 

Please combine the "methods" and "findings" subsections of your abstract. The final sentence of the new combined subsection should quote 2-3 of the study's main limitations. 

Please indicate in the abstract that the studies were done in high-income countries (you may be able to list these, as the number is modest). Also, please quote the range of study sizes. 

Please mention study quality in your abstract.

Please begin the "conclusions" subsection of your abstract with "In this study, we found that ..." or similar. 

After the abstract, we ask you to include a new and accessible "author summary" section in non-identical prose. You may find it helpful to consult one or two recent research papers published in PLOS Medicine to get a sense of the preferred style. 

Please add a short sentence to the methods section to note that ethics approval was not needed. 

Please restructure the Discussion section of your main text so that there is a discrete paragraph discussing limitations. 

Surely confidence intervals should be presented as "95% CI"?

Please quote exact p values or p<0.001, throughout the ms, unless there is a specific statistical reason for presenting smaller exact values. 

Throughout the paper, please adapt reference call-outs to the following style: "... utilization [1-4].".

In your reference list, please abbreviate journal names consistently (e.g., "Lancet" for reference 1).

Please adapt the attached PRISMA checklist so that individual items are referred to by section (e.g., "Methods") and paragraph number rather than by page or line numbers, as the latter generally change in the event of publication. Please refer to the checklist in the main text. 

Comments from the reviewers:

*** Reviewer #1 (methodological): 

The authors report the findings a systematic review which has determined, from observational cohort studies, that patients with severe mental illness are more likely to use non-psychiatric health services. They have performed several random-effects meta-analyses quantifying the effect size of admissions, hospital stay length, hospital readmissions and A&E. 

Although the aims were framed about health service usage, including use of primary care services, the meta-analysis outcomes really are focussed on secondary care admissions and stays, given a smaller number of studies exploring primary care utilisation. Prior to consideration for publication, there area number of issues that should be addressed. In particular, the search is over a year old and it would extremely advisable to update the review as it's likely that new studies in the past year may fit the eligibility criteria. Overall the review has merit but some specific issues need to be addressed below:

Specific comments:

Abstract - please state the type of studies included (i.e. observational cohort, case control, RCT etc)

Abstract - It's important to quantify the heterogeneity in the abstract to give the readers an idea about the variation in the study outcomes between studies to help with the interpretation on how 

Introduction - The authors rationalise that the evidence to date suggests psychiatric comorbidity is associated with increased utilisation based on two previous reviews (6) and (7), but the impact of severe mental illness has on health care utilisation has not yet been assessed. This is only partially correct. The most recent review (ref 7) did not explicitly preclude patients with SMI and in fact does in include a number of studies patients with SMI. Though the case could be made the case that this review is more focussed, specifically just on SMI. 

Search Strategy: The searches carried out are more than a year old. The authors state the search strategy was conducted between 26th of Oct - 2nd of Nov 2018. it would be extremely advisable to update the search. For instance - this study which was published Dec 2018 could warrant inclusion: https://www.ncbi.nlm.nih.gov/pmc/articles/PMC6236443/ and another on an RCT published Nov 12 2018 may fit the criteria as well https://bmcpsychiatry.biomedcentral.com/articles/10.1186/s12888-018-1941-2

 Definition of SMI: How did the authors deal with studies which may have had overlap between illnesses with psychosis and major depressive disorders as often are present in the same individual. If the case definition didn't preclude major depressive disorder - according to the way the study population inclusion is written - these studies would be excluded as the authors state "Studies that included MDD in their definition of SMI were excluded unless results were presented separately for each SMI subtype"

Primary outcome: Health service utilisation was the primary outcome but the authors need to describe the unit of measurement here (e.g. number of admissions, length of stay). 

Study types: Have the authors considered including health economic modelling studies. Health service utilisation is often times seen in health economic modelling papers. Any particular reason this may not have been explored, considering most other study designs with included. Linking back to the introduction argument made that these results "might be used by policy-makers and health economists to improve delivery of integrated care". It would seem plausible to also include health economic papers (which are largely based on observational data to parameterise models). 

Study age inclusion rationale: I'm not sure if I really understand the rationale for the likelihood of an index condition occurring or not as the primary limiting factor for the age cut-off. I would have thought that authors could have just stated that the limiting factor is that younger teenagers/children may have quite a different rate of health care utilisation due to having fewer co-morbidities related to chronic conditions which occur more in older adults. 

Data extraction and quality assessment: CCI and ECI index - did the requirement for covariate adjustment have to be a composite index? Or could multiple covariate adjustment for co-morbidities be sufficient and/or if studies used propensity scores? It's not exactly clear here?

The meta-analysis performed where possible using the random-effects model for studies of this nature. Expectedly, there is significant amounts of heterogeneity between (Figure 2 - 100%; Figure 3 - 100%; Figure 4 - 85%; Figure 5 - 99%). High heterogeneity does not invalidate the results but in a large review of this nature - this does mean the headline figures need extreme caution in interpretation - given > 85% I^2. Hence why it's important to present he I^2 in the abstract

*** Reviewer #2: 

PMEDICINE-D-20-00254R1: The impact of severe mental illness on health service utilisation for non-psychiatric medical disorders: A systematic review and meta-analysis

In this manuscript, the authors aim to assess the specific impact of Severe Mental Illness (SMI) on the use of inpatient, emergency, and primary care services for non-psychiatric medical disorders. Physical co-morbidity in individuals with SMI is an extremely important area of work and through this extremely well-written draft, the authors have highlighted the importance of system-wide integration of mental and physical health services. There are few minor suggestions. 

1. It is very appropriately mentioned that disparities exist in the provision of healthcare for individuals with SMI. However, the findings of this review suggest that there is increased use of general medical services including primary health services. The likely explanation provided is the worsening of the physical symptoms by the time they are presented to health services. Although this is not the primary research question, it is possible to assess this speculation based on the reviewed data? Did individuals with SMI had more severe physical illness compared to controls? Given that this physical illness and co-morbidity were adjusted in analysis, it will be good to comment on this further. 

2. Length of Stay was 0.74 days more in individuals with SMI. The meta-analysis findings suggest that this was highly significant. Please comment on how significant is this from clinical management point of view? Is this difference quite huge? Similar data from other areas of health research will be useful. 

3. All the studies are from high income countries and it will be interesting to see the utilization of medical services in low resource settings/low income countries. It will be good to include this in discussion.

4. It is possible to quickly refresh your search and include any new studies after Nov 2018? It won't possibly be a huge effort, but will ensure that the results presented are absolutely up-to-date. 

5. The case for system-wide integrated care needs to be further strengthened. A reader might get an impression that the service utilization in primary care, in-patient admissions, LOS and re-admission is already increased, hence the care is already integrated. 

*** Reviewer #3: 

Please see my comments attached. Overall, I need more clarity regarding the methods.

*** Reviewer #4: 

Thanks you for you interesting and timely work. The authors are to be commended for their contribution in building a case for system-wide integration of medical and mental healthcare. 

My main comments are as follows:

-Why do the authors stick to the method of meta-analysis when heterogeneity is large and narrative review would suffice? I would suggest changing this. 

-The authors didn't include a discussion of the pros and cons of the Newcastle-Ottawa Scale, please amend.

-In the discussion, the section on factors that might explain why patients with SMI are using non-psychiatric healthcare serivices more thant those without is insufficiently substantiated in my opinion. The number of hypotheses to explain the differences found is limited, as well as the embedding in the literature. This must and can really be better and more complete, certainly considering the main purpose of the review.

***

[LINK]

---

## [Decision Letter · Decision Letter 2]

1 Jul 2020

Dear Dr. Ronaldson,

Thank you very much for re-submitting your manuscript "Severe mental illness and health service utilisation for non-psychiatric medical disorders: A systematic review and meta-analysis" (PMEDICINE-D-20-00254R2) for consideration at PLOS Medicine.

I have discussed the paper with editorial colleagues and it was also seen again by one reviewer. I am pleased to tell you that, provided the remaining editorial and production issues are fully dealt with, we expect to be able to accept the paper for publication in the journal.

[LINK]

Please let me know if you have any questions. Otherwise, we look forward to receiving the revised manuscript shortly. 

Sincerely,

Richard Turner, PhD

rturner@plos.org

Requests from Editors:

We suggest noting the date of the search update in the abstract.

We suggest deleting the three country names from the abstract, although as an alternative you may wish to state the number of studies carried out in the United States. 

You mention in the abstract that study quality was assessed. Please briefly state what was found. 

Please revisit the end of the "methods and findings" section of your abstract. The final sentence of this subsection should begin "Study limitations include ..." or similar; you may wish to abbreviate and combine the two sentences currently at the end of this subsection to create this single final sentence. 

Please delete the two sentences at lines 108-112, or move the text to the discussion section. 

At line 301 and other instances, we think that "Higgins' " would be the appropriate form.

Noting "p=0.0002" at line 308, we generally ask that "p<0.001" is written unless there are specific statistical reasons to the contrary. Please also revisit the figures with this issue in mind. 

At line 357, please make that "... to our knowledge the first" or similar. 

The first paragraph of the Discussion section, which should summarize the study's findings, is quite brief, and we suggest adding an additional sentence, say, to make this more explicit. 

At line 462, you may wish to adapt this to "low- and middle-income countries".

At the end of the main text, please remove the statements on funding, conflict of interest, ethics, data availability and consent for publication. Information on three of these items will appear in the metadata upon publication, via information provided in the submission form. 

Please remove the tracking from the attached PRISMA checklist.

Comments from Reviewers:

*** Reviewer #2: 

Thank you for addressing all the comments.

***

[LINK]

---

## [Editor Report · Decision Letter 3]

10 Aug 2020

Dear Dr Ronaldson, 

On behalf of my colleagues and the academic editor, Dr. Rahul Shidhaye, I am delighted to inform you that your manuscript entitled "Severe mental illness and health service utilisation for non-psychiatric medical disorders: A systematic review and meta-analysis" (PMEDICINE-D-20-00254R3) has been accepted for publication in PLOS Medicine. 

PRODUCTION PROCESS

PRESS

PROFILE INFORMATION

Thank you again for submitting the manuscript to PLOS Medicine. We look forward to publishing it. 

Best wishes, 

Richard Turner, PhD

Senior Editor 

PLOS Medicine

plosmedicine.org